# CurvZO: Adaptive Curvature-Guided Sparse Zeroth-Order Optimization for Efficient LLM Fine-Tuning

**Shuo Wang** [1]   **Ziyu Chen** [2]   **Ming Tang** [1]

## Abstract

Fine-tuning large language models (LLMs) with backpropagation achieves high performance but incurs substantial memory overhead, limiting scalability on resource-constrained hardware. Zeroth-order (ZO) optimization provides a memory-efficient alternative by relying solely on forward passes, yet it typically suffers from slow or unstable convergence due to high-variance gradient estimates. Sparse ZO updates partially address this issue by perturbing only a subset of parameters, but their effectiveness hinges on selecting informative parameters, which is challenging in ZO optimization because each query yields only scalar feedback. We propose **Adaptive Curvature-Guided Sparse Zeroth-Order Optimization (CurvZO)**, which tracks curvature signals online from scalar ZO feedback and leverages these signals to construct a parameter-wise sampling distribution for selecting coordinates at each update, reducing the variance of the sparse ZO gradient estimator. Moreover, CurvZO dynamically adapts the perturbation budget to the evolving curvature signal distribution, yielding sparse ZO updates that remain both focused and sufficiently exploratory. Extensive experiments on OPT and Llama across diverse NLP tasks show that CurvZO consistently improves fine-tuning performance and reduces training time over ZO baselines. It improves accuracy by up to 4.4 points and achieves up to a $2\times$ speedup, while preserving memory efficiency.

[1]Department of Computer Science and Engineering, Southern University of Science and Technology, Shenzhen, China. [2]Department of Mathematics, Southern University of Science and Technology, Shenzhen, China. Correspondence to: Ming Tang <tangm3@sustech.edu.cn>.

*Proceedings of the $43^{rd}$ International Conference on Machine Learning*, Seoul, South Korea. PMLR 306, 2026. Copyright 2026 by the author(s).

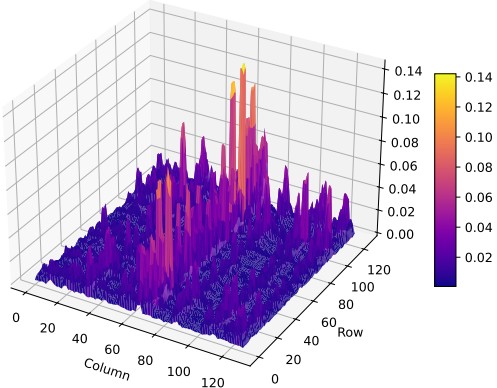

*Figure 1.* Visualization of anisotropic local curvature in the attention output weights of OPT-6.7B. The $x$- and $y$-axes index the columns and rows of the weight matrix, while the $z$-axis shows curvature magnitude approximated via the diagonal Fisher information (a standard local-curvature surrogate in neural networks).

## 1. Introduction

Fine-tuning large language models (LLMs) via backpropagation is a widely adopted strategy that achieves strong performance across a variety of tasks. However, it imposes significant memory overhead due to the large number of parameters, which limits its scalability on resource-constrained hardware. As model sizes continue to grow rapidly, often by orders of magnitude every few years, the gap between memory requirements and available hardware capacity becomes increasingly pronounced (Hoffmann et al., 2022; Kaplan et al., 2020). This trend intensifies the so-called memory wall (Gholami et al., 2024) and poses challenges for both training and deployment (Zeng et al., 2024; Hur et al., 2023).

To address these limitations, zeroth-order (ZO) optimization has emerged as a memory-efficient alternative for fine-tuning LLMs (Tan et al., 2026; Liu et al., 2026; Zhao et al., 2025; Zhang et al., 2024). ZO methods estimate gradients using only forward passes, avoiding backpropagation and largely eliminating the memory cost of storing activations, gradients, and optimizer states. Among them, MeZO (Malladi et al., 2023) further reduces memory by re-generating perturbations from a saved random seed, so training memory becomes close to inference, yielding up to $12\times$ savings

in practice.

However, ZO optimization often suffers from unstable convergence and lower accuracy compared to first-order (FO) methods that rely on backpropagation for gradient computation, largely due to noise in ZO gradient estimates (Liu et al., 2026). Prior studies have indicated that this noise scales with the dimensionality of the optimized parameters, leading to further degradation as dimensionality increases (Liu et al., 2026; Yu et al., 2025; Guo et al., 2025). Although prior works attempt to mitigate this issue by reducing the number of parameters involved in ZO fine-tuning, they do not provide a principled way to identify which subset of parameters should be perturbed. This limitation arises because the accessible information in the ZO setting is fundamentally restricted to scalar responses from random perturbations. To address this, existing methods incorporate additional information or pre-computed statistics. For instance, SensZOQ (Guo et al., 2025) constructs a sparsity pattern using the empirical Fisher information matrix, computed offline before fine-tuning. Such reliance on additional information or pre-computed statistics introduces non-negligible computational overhead, thereby undermining the primary advantage of ZO methods: memory efficiency and simplicity.

This naturally leads to the following question: *Can we design a plug-and-play mechanism that selectively allocates perturbations to the most informative parameters, thereby improving ZO fine-tuning without relying on pre-computed statistics or changes to the existing framework?*

Recent studies have shown that the loss landscape of LLMs can exhibit highly anisotropic curvature (Zhao et al., 2025). In particular, local curvature magnitudes vary substantially across parameters (Figure 1), providing a proxy for local sensitivity. This observation suggests that an effective ZO optimization algorithm could leverage curvature information to prioritize informative parameter subsets and guide perturbations toward higher-curvature directions. However, achieving this remains challenging in the ZO setting, where each query yields only a scalar response, and neither gradients nor curvature can be directly accessed.

In this work, we propose **Adaptive Curvature-Guided Sparse Zeroth-Order Optimization (CurvZO)**, a framework that tracks curvature signals online from scalar responses and leverages them to construct a sampling distribution for sparse perturbations. CurvZO reduces the variance of the sparse ZO gradient estimator by perturbing parameters with larger local curvature signals more frequently. Moreover, we develop an adaptive budget mechanism that dynamically adjusts the number of perturbed parameters according to the evolving curvature signal distribution, yielding sparse ZO updates that remain both focused and sufficiently exploratory.

The summary of our contributions is as follows:

- **Online Curvature Signal Tracking.** We propose a curvature score to track curvature signals online from the scalar responses of ZO updates, which captures the local geometry of the loss landscape and provides guidance for selective perturbations in sparse ZO.

- **Curvature-Guided Sparse ZO Fine-Tuning.** Leveraging online curvature scores, we design a curvature-guided sampling rule that assigns higher sampling probabilities to parameters with larger scores. This rule concentrates perturbations on higher-curvature directions, thereby reducing the variance of the sparse ZO gradient estimator.

- **Adaptive Budget Selection.** We introduce an adaptive budget mechanism that adjusts the number of perturbed parameters based on the evolving curvature score distribution, yielding sparse ZO updates that remain both focused and sufficiently exploratory.

- **Empirical Evaluation.** We evaluate CurvZO on OPT and Llama models across diverse NLP tasks. CurvZO consistently outperforms ZO baselines and converges faster, improving accuracy by up to 4.4 points and reducing GPU hours by up to $2\times$, while preserving ZO-level memory efficiency.

**Conflict of Interest Disclosure.** The authors declare that they have no financial conflicts of interest related to this work.

## 2. Preliminaries

### 2.1. Zeroth-Order Optimization

ZO optimization has recently attracted significant attention in machine learning (Verma et al., 2023; Wang et al., 2022; Gu et al., 2021). It estimates gradients using only a few forward passes, without requiring backpropagation. Crucially, ZO eliminates the need to store memory-intensive components required in FO training, such as intermediate activations, backward gradients, and optimizer states, thereby substantially reducing GPU memory consumption.

A classical gradient estimator used in ZO methods is the Simultaneous Perturbation Stochastic Approximation (SPSA) (Spall, 2002). Each ZO query yields a scalar response via finite-difference approximation:

$$\Delta = \frac{\mathcal{L}(\boldsymbol{\omega} + \epsilon\boldsymbol{z}; \mathcal{B}) - \mathcal{L}(\boldsymbol{\omega} - \epsilon\boldsymbol{z}; \mathcal{B})}{2\epsilon}, \qquad (1)$$

where $\boldsymbol{\omega} \in \mathbb{R}^d$ denotes the model parameters, and $\mathcal{L}(\boldsymbol{\omega}; \mathcal{B})$ is the loss evaluated on a minibatch $\mathcal{B}$ of size $B$, uniformly

sampled from the training dataset $\mathcal{D} = \{(x_i, y_i)\}_{i=1}^{|\mathcal{D}|}$. Meanwhile, $z \in \mathbb{R}^d$ is a random perturbation sampled from $\mathcal{N}(0, I_d)$, and $\epsilon$ is the perturbation scale. The ZO gradient estimator is given by

$$\hat{\nabla}\mathcal{L}(\boldsymbol{\omega}; \mathcal{B}) = \Delta z. \qquad (2)$$

Then, this estimator can be plugged into standard FO optimizers. For example, ZO-SGD updates the parameters as

$$\boldsymbol{\omega}^{t+1} = \boldsymbol{\omega}^t - \eta^t \hat{\nabla}\mathcal{L}(\boldsymbol{\omega}^t; \mathcal{B}), \qquad (3)$$

where $\eta^t > 0$ is the learning rate at iteration $t$.

## 2.2. Sparsity in ZO

Existing studies have established that ZO fine-tuning performance degrades as the dimensionality of the optimized parameters increases (Liu et al., 2026; Yu et al., 2025). To mitigate this, recent works enforce sparsity by perturbing only a subset of parameters per update. Specifically, they construct a random binary mask $m \in \{0, 1\}^d$ and apply it to Gaussian noise $z \sim \mathcal{N}(0, I_d)$ to obtain a sparse perturbation $\hat{z} = m \odot z$. The gradient is then estimated via SPSA with $\hat{z}$, i.e., by replacing $z$ with $\hat{z}$ in Eqs. (1) and (2).

Theoretical analyses of ZO with sparse perturbations show that sparsity can accelerate convergence under smoothness assumptions (Liu et al., 2026; Guo et al., 2025). Motivated by this, Sparse-MeZO (Liu et al., 2026) restricts perturbations to small-magnitude weights. However, the perturbation injected into each parameter is $z_i \sim \mathcal{N}(0, 1)$, whose absolute scale is independent of the parameter magnitude. Therefore, perturbing small-magnitude parameters can cause disproportionately large changes. This increases the variance of the resulting ZO gradient estimates, which can result in less reliable and suboptimal update directions. In addition, SubZero (Yu et al., 2025) achieves sparsity by imposing a low-rank structure on the perturbations. Although both Sparse-MeZO and SubZero recognize that introducing sparsity can improve the efficiency of ZO optimization, they lack an effective mechanism to identify the optimal subset of parameters to perturb. Their sparsity patterns are either predefined or randomly assigned, rather than adaptively determined according to the importance or sensitivity of parameters, which may limit their overall effectiveness.

Moreover, SensZOQ (Guo et al., 2025) constructs a static sparsity pattern using empirical Fisher information computed during pre-training. However, this approach relies on pre-computed, gradient-based statistics, which incur non-negligible computational overhead.

## 3. CurvZO

We consider the sparse ZO setting, where each update perturbs only a subset of parameters. Given a limited perturba-

tion budget $B$, a key challenge is to allocate perturbations to the most informative parameters, so that each query yields the largest possible optimization progress. Consequently, the selection of parameters to perturb is central to the efficiency of sparse ZO optimization.

To this end, we introduce a binary mask $m \in \{0, 1\}^d$, where $m_i = 1$ indicates that the $i$-th parameter is selected for perturbation and $m_i = 0$ otherwise. We generate $m$ by sampling each entry independently as $m_i \sim \text{Bernoulli}(\pi_i)$, where $\pi_i \in (0, 1]$ denotes the sampling probability of the $i$-th parameter. Given $z \sim \mathcal{N}(0, I_d)$, we construct a sparse perturbation direction $v = m \odot z$.

The challenge is to assign sampling probabilities $\boldsymbol{\pi} = (\pi_1, \dots, \pi_d)$ over parameter coordinates so as to prioritize the most informative ones. However, ZO feedback offers little direct guidance for setting these probabilities: each query yields only a scalar response $\Delta$, with no explicit coordinate-wise signal indicating which coordinates are most beneficial to perturb. On the other hand, curvature serves as a proxy for local sensitivity, so it is natural to construct $\boldsymbol{\pi}$ so that parameters with higher local curvature receive larger sampling probabilities (Zhao et al., 2025). However, curvature is not directly observable in ZO optimization, so constructing $\boldsymbol{\pi}$ from scalar feedback alone is nontrivial.

We introduce **Adaptive Curvature-Guided Sparse Zeroth-Order Optimization (CurvZO)**, a framework that tracks curvature signals online and uses them to guide sparse ZO updates. At each iteration, CurvZO tracks per-parameter curvature signals from the scalar response $\Delta$. It then uses these signals to construct a sampling distribution $\boldsymbol{\pi}$ over parameters and samples a sparse perturbation mask $m$ accordingly, so that parameters with larger curvature signals are perturbed more frequently. CurvZO also adapts the perturbation budget $B$ to align the sparsity level with the evolving curvature signal distribution.

### 3.1. Online Curvature Signal Tracking

In LLMs, curvature is often approximated using Fisher information matrix (Kirkpatrick et al., 2017), or more commonly its diagonal. For models trained with the cross-entropy loss $\mathcal{L}$, the $i$-th diagonal entry is defined as:

$$F_{ii}(\boldsymbol{\omega}) = \mathbb{E}_{(x,y) \sim P_{\text{data}}} \left[ \left( \frac{\partial}{\partial \omega_i} \mathcal{L}(\boldsymbol{\omega}; x, y) \right)^2 \right]. \qquad (4)$$

However, in ZO optimization, the only available information per query is the perturbation direction $v$ and the scalar response $\Delta$, making gradient-based curvature estimation infeasible. Therefore, we track an online curvature signal for Eq. (4) from $v$ and $\Delta$.

**Curvature Score.** We define a curvature score and use it as the curvature signal, given by

$$s_i \triangleq \Delta^2 v_i^2. \quad (5)$$

To show that $s_i$ tracks a meaningful curvature signal, we take expectation over the perturbation randomness (i.e., $\boldsymbol{m}$ and $\boldsymbol{z}$), which yields

$$\mathbb{E}_{\boldsymbol{m},\boldsymbol{z}}[s_i] = \underbrace{\pi_i \sum_{j=1}^{d} \pi_j g_j^2}_{\text{shared term}} + \underbrace{\pi_i(3-\pi_i)g_i^2}_{\text{signal term}} + \mathcal{O}(\epsilon^2), \quad (6)$$

where $g_i$ is the true gradient of parameter $i$. The derivation is provided in Appendix A. In Eq. (6), the signal term contains $g_i^2$, which is widely used as a stochastic proxy for the diagonal of the Fisher information matrix. Therefore, $s_i$ can track a meaningful curvature signal, and we use it to construct the sampling distribution $\boldsymbol{\pi}$ and guide sparse perturbations. While an unbiased variant of $s_i$ can eliminate the shared term in Eq. (6) in expectation, it typically incurs much higher variance and can be unstable. Details are provided in Appendix B.

**Stabilizing Curvature Scores.** The raw curvature score $s_i$ can be noisy due to (i) the high variance of $v_i^2$ induced by Bernoulli sparsity and Gaussian perturbations, and (ii) temporal variability in the ZO response $\Delta$. We address these two sources via normalization and temporal smoothing.

To reduce the variance induced by $v_i^2$, we normalize the per-coordinate contribution by the total perturbation energy $\sum_j v_j^2$. This yields a curvature score that is less sensitive to the absolute scale of individual $v_i^2$, improving practical stability. Specifically, we define the *normalized curvature score* for parameter $i$ as:

$$\tilde{s}_i = \frac{v_i^2}{\sum_j v_j^2} \Delta^2. \quad (7)$$

To further mitigate temporal variability, we apply exponential moving average (EMA), forming the smoothed curvature scores $\mathcal{S}^t = \{S_i^t\}_{i=1}^d$ at iteration $t$ via

$$S_i^t \leftarrow (1-\beta)S_i^{t-1} + \beta \tilde{s}_i^t, \quad (8)$$

where $\beta \in (0,1]$ controls the degree of smoothing, with smaller values yielding smoother and more stable curvature scores.

## 3.2. Curvature-Guided Sparse Zeroth-Order Optimization

We introduce curvature-guided sparse ZO optimization, a framework that uses curvature scores $\mathcal{S}^t$ to construct a sampling distribution $\boldsymbol{\pi}^t = \{\pi_i^t\}_{i=1}^d$ for sparse perturbations.

Specifically, we first show that the naive gradient estimator $\hat{\boldsymbol{g}} = \Delta \boldsymbol{v}$ is biased under Bernoulli-based sparse masking (where $m_i \sim \text{Bernoulli}(\pi_i^t)$ and $\boldsymbol{v} = \boldsymbol{m} \odot \boldsymbol{z}$), and correct it using Horvitz–Thompson reweighting, yielding an unbiased estimator $\tilde{g}_i = \Delta \frac{v_i}{\pi_i^t}$. We then derive $\boldsymbol{\pi}^t$ by minimizing an upper bound on the variance of the corrected estimator under a fixed perturbation budget $B$, which yields the curvature-aware sampling rule $\pi_i^t \propto \sqrt{S_i^t}$.

**Bias Correction for the Gradient Estimator.** Given $\boldsymbol{z} \sim \mathcal{N}(\boldsymbol{0}, \boldsymbol{I}_d)$, we sample a binary mask $\boldsymbol{m}$ with independent entries $m_i \sim \text{Bernoulli}(\pi_i)$ and form the sparse perturbation direction $\boldsymbol{v} = \boldsymbol{m} \odot \boldsymbol{z}$.

The naive gradient estimator $\hat{\boldsymbol{g}} \triangleq \Delta \boldsymbol{v}$ is biased. In particular, for each coordinate $i$, we have

$$\mathbb{E}_{\boldsymbol{m},\boldsymbol{z}}[\hat{g}_i] = \mathbb{E}_{\boldsymbol{m},\boldsymbol{z}}[(\boldsymbol{g}^T(\boldsymbol{m} \odot \boldsymbol{z}))m_i z_i] = g_i \pi_i. \quad (9)$$

Therefore, aggregating across coordinates yields

$$\mathbb{E}_{\boldsymbol{m},\boldsymbol{z}}[\hat{\boldsymbol{g}}] = \mathbb{E}_{\boldsymbol{m},\boldsymbol{z}}[\Delta \boldsymbol{v}] = \text{diag}(\boldsymbol{\pi})\,\boldsymbol{g} + \mathcal{O}(\epsilon^2). \quad (10)$$

Hence, $\hat{\boldsymbol{g}}$ is unbiased only when $\pi_i = 1$ for all $i$; otherwise, each coordinate is attenuated in expectation by its sampling probability $\pi_i$. When $\pi_i$ varies across coordinates, the expected update becomes a coordinate-wise rescaling of the true gradient, which can distort the optimization dynamics (e.g., slowing convergence and complicating standard convergence analyses).

To remove this sampling-induced bias, we adopt the Horvitz–Thompson correction (Särndal et al., 2003), i.e., an inverse-probability reweighting that rescales the $i$-th component of the gradient estimate by $1/\pi_i$.

**Definition 3.1.** Given the sparse perturbation $\boldsymbol{v} = \boldsymbol{m} \odot \boldsymbol{z}$ and the two-point finite-difference response

$$\Delta \triangleq \frac{\mathcal{L}(\boldsymbol{\omega} + \epsilon \boldsymbol{v}; \mathcal{B}) - \mathcal{L}(\boldsymbol{\omega} - \epsilon \boldsymbol{v}; \mathcal{B})}{2\epsilon}, \quad \epsilon > 0, \quad (11)$$

we define the gradient estimator $\tilde{\boldsymbol{g}} \triangleq (\tilde{g}_i)_{i=1}^d$ with

$$\tilde{g}_i \triangleq \Delta \frac{v_i}{\pi_i}. \quad (12)$$

We next formalize the unbiasedness of $\tilde{\boldsymbol{g}}$.

**Proposition 3.2** (Unbiasedness)**.** *The gradient estimator defined in Definition 3.1 is unbiased, i.e.,*

$$\mathbb{E}_{\boldsymbol{m},\boldsymbol{z}}[\tilde{\boldsymbol{g}}] = \boldsymbol{g} + \mathcal{O}(\epsilon^2), \quad (13)$$

*where $\mathcal{O}(\epsilon^2)$ denotes the standard second-order bias in ZO gradient estimation.*

The proof is provided in Appendix C. This correction eliminates the sampling bias induced by Bernoulli-based sparse masking, making the gradient estimator unbiased up to the standard $\mathcal{O}(\epsilon^2)$. This unbiasedness is essential for CurvZO to optimize the intended objective and converge to a stationary point.

**Variance-Minimizing Sampling Distribution.** Since the unbiased gradient estimator in Definition 3.1 depends on $\boldsymbol{\pi}^t$ via the $1/\pi_i^t$ reweighting, we design $\boldsymbol{\pi}^t$ by minimizing an upper bound on the total variance subject to a sampling budget $B = \sum_i \pi_i^t$.

We begin by establishing an upper bound on the variance of the unbiased gradient estimator.

**Proposition 3.3** (Variance of the Unbiased Gradient Estimator). *For the ZO estimator in Definition 3.1, the coordinate-wise variance satisfies*

$$\operatorname{Var}(\tilde{g}_i^t) \leq C \frac{F_{ii}}{\pi_i^t}, \tag{14}$$

*for some constant $C > 0$, where $F_{ii}$ denotes the $i$-th diagonal entry of the Fisher information matrix.*

This bound suggests constructing $\boldsymbol{\pi}^t$ to minimize the total variance upper bound, yielding the following problem (P1):

$$\text{(P1)} \quad \boldsymbol{\pi}^{t\star} \in \arg\min_{\boldsymbol{\pi}^t} \sum_{i=1}^{d} \frac{F_{ii}}{\pi_i^t}$$

$$\text{s.t.} \sum_{i=1}^{d} \pi_i^t = B, \tag{15}$$

$$0 \leq \pi_i^t \leq 1, \ \forall i.$$

**Proposition 3.4** (Variance-Minimizing Sampling Distribution). *Let $\boldsymbol{\pi}^{t\star} = \{\pi_i^{t\star}\}_{i=1}^{d}$ be an optimal solution to Problem (P1). It holds that*

$$\pi_i^{t\star} \propto \sqrt{F_{ii}}. \tag{16}$$

*The probabilities are clipped at $1$ to satisfy $\pi_i^t \leq 1$.*

Recall that in practice, the Fisher diagonals $F_{ii}$ are not observable in ZO training. We use the curvature scores $\mathcal{S}^t = \{S_i^t\}_{i=1}^{d}$ and adopt a plug-in rule:

$$\pi_i^{t\star} \propto \sqrt{S_i^t}. \tag{17}$$

Intuitively, the variance bound scales as $F_{ii}/\pi_i$; if a high-curvature coordinate is sampled rarely (small $\pi_i$), its variance term becomes large. Thus, assigning larger $\pi_i$ to coordinates with larger curvature scores reduces the overall estimator variance. The proofs of Propositions 3.3 and 3.4 are provided in Appendices D and E, respectively.

### 3.3. Adaptive Budget Selection

In ZO optimization, curvature scores evolve over training. Early iterations often yield noisy and flat curvature scores, for which a larger sampling budget helps maintain exploration and stable updates; later, as the curvature score distribution becomes sharper, a smaller budget typically suffices.

Accordingly, a fixed budget can be suboptimal, and we adapt the sampling budget $B$ dynamically using two statistics of the score distribution: *effective support size* and *sharpness*.

**Effective Support Size.** Motivated by the variance-minimizing sampling rule $\pi_i^\star \propto \sqrt{S_i}$, we define the *effective support size* as the effective number of coordinates contributing to the score distribution:

$$d_{\text{eff}} = \frac{\left(\sum_i \sqrt{S_i}\right)^2}{\sum_i S_i} \in [1, d]. \tag{18}$$

A smaller $d_{\text{eff}}$ indicates that scores concentrate on fewer coordinates, so a smaller sampling budget is typically sufficient; conversely, when the scores are more broadly distributed across coordinates, $d_{\text{eff}}$ approaches $d$ (the total number of parameter coordinates), suggesting a larger budget.

**Score Sharpness.** To quantify the sharpness of the curvature score distribution, we compute the normalized Shannon entropy (Shannon, 1948; Gal & Ghahramani, 2016):

$$H = -\frac{1}{\log d} \sum_{i=1}^{d} p_i \log p_i \in [0, 1], \tag{19}$$

where $p_i \triangleq \frac{\sqrt{S_i}}{\sum_{j=1}^{d} \sqrt{S_j}}$ is the normalized score distribution induced by the variance-minimizing sampling rule $\pi_i \propto \sqrt{S_i}$. A larger $H$ indicates a more uniform (less decisive) distribution, motivating a larger budget; a smaller $H$ indicates a sharper distribution, allowing a smaller budget.

**Adaptive Budget Selection.** We set the sampling budget $B$ by combining effective support size and sharpness:

$$B = B_{\min} + (B_{\max} - B_{\min}) \left( \alpha \frac{d_{\text{eff}}}{d} + (1-\alpha)H \right), \tag{20}$$

where $\alpha \in [0, 1]$ controls the trade-off between the two measures.

Intuitively, $d_{\text{eff}}/d$ increases as the scores become less concentrated (i.e., more coordinates contribute), and $H$ increases as the induced distribution becomes more uniform; both indicate that a larger budget is needed. Conversely, when the scores are concentrated on fewer coordinates and the distribution is sharper, a smaller budget suffices.

### 3.4. Convergence Analysis

In this section, we present a convergence analysis of the proposed CurvZO. The analysis is performed on the smoothed objective $\mathcal{L}_\pi(\boldsymbol{\omega}) = \mathbb{E}_{\boldsymbol{m},\boldsymbol{z}}[\mathcal{L}(\boldsymbol{\omega} + \epsilon \boldsymbol{v})]$, where $\boldsymbol{v} = \boldsymbol{m} \odot \boldsymbol{z}$ is the sparse perturbation. Specifically, we (i) establish a standard descent inequality for the smoothed loss, (ii) leverage curvature-guided sampling to bound the second

moment of the sparse gradient estimator, and (iii) relate the smoothed gradient back to the true gradient of the original objective $\mathcal{L}(\boldsymbol{\omega})$. Our main result shows that, under a fixed stepsize and a variance-minimizing sampling budget satisfying $B = \sum_i \pi_i$, CurvZO achieves an $\mathcal{O}(1/T)$ convergence rate up to a variance term depending on $B$ and a small smoothing bias controlled by $\epsilon$.

Firstly, we assume that the loss function $\mathcal{L}(\boldsymbol{\omega})$ is $L$-smooth.

**Assumption 3.5** (Lipschitz Continuous). The loss function $\mathcal{L}(\boldsymbol{\omega})$ has an $L$-Lipschitz continuous gradient, i.e.,

$$\|\nabla\mathcal{L}(\boldsymbol{\omega}) - \nabla\mathcal{L}(\boldsymbol{\omega}')\| \leq L \|\boldsymbol{\omega} - \boldsymbol{\omega}'\|, \quad (21)$$

where $\nabla\mathcal{L}(\boldsymbol{\omega})$ denotes the true first-order gradient of $\mathcal{L}$ at $\boldsymbol{\omega}$, and $L > 0$ is the smoothness constant.

Note that $\mathcal{L}_\pi$ inherits $L$-smoothness from $\mathcal{L}$: since $\mathcal{L}_\pi(\boldsymbol{\omega}) = \mathbb{E}_{\boldsymbol{m},\boldsymbol{z}}[\mathcal{L}(\boldsymbol{\omega} + \epsilon\boldsymbol{v})]$ averages $\mathcal{L}$ over random perturbations in a neighborhood of $\boldsymbol{\omega}$, $\mathcal{L}_\pi$ is also $L$-smooth.

Under Assumption 3.5, for any $\boldsymbol{\omega}, \boldsymbol{\omega}' \in \mathbb{R}^d$, we have

$$\mathcal{L}(\boldsymbol{\omega}') \leq \mathcal{L}(\boldsymbol{\omega}) + \langle \nabla\mathcal{L}(\boldsymbol{\omega}), \boldsymbol{\omega}' - \boldsymbol{\omega} \rangle + \frac{L}{2} \|\boldsymbol{\omega}' - \boldsymbol{\omega}\|^2. \quad (22)$$

In particular, by setting $\boldsymbol{\omega} = \boldsymbol{\omega}_t$ and $\boldsymbol{\omega}' = \boldsymbol{\omega}_{t+1} = \boldsymbol{\omega}_t - \eta\tilde{\boldsymbol{g}}^t$, Eq. (22) yields a one-step descent inequality for $\mathcal{L}_\pi$, which will be used in the subsequent convergence analysis.

Next, to control the difference between the gradient of the smoothed loss and that of the original loss, we introduce the following higher-order regularity assumption.

**Assumption 3.6** (Hessian Lipschitz Continuity). The loss function $\mathcal{L}(\boldsymbol{\omega})$ is three–times continuously differentiable and its Hessian $\nabla^2\mathcal{L}(\boldsymbol{\omega})$ is $\rho$-Lipschitz continuous, i.e.,

$$\|\nabla^2\mathcal{L}(\boldsymbol{\omega}) - \nabla^2\mathcal{L}(\boldsymbol{\omega}')\| \leq \rho \|\boldsymbol{\omega} - \boldsymbol{\omega}'\|, \forall \boldsymbol{\omega}, \boldsymbol{\omega}' \in \mathbb{R}^d. \quad (23)$$

This assumption is standard in analyses of randomized smoothing and ZO methods, as it controls the third-order remainder in Taylor expansions.

Under this assumption, the following lemma bounds the original gradient norm in terms of the smoothed gradient norm.

**Lemma 3.7** (Bounding $\|\nabla\mathcal{L}\|$ by $\|\nabla\mathcal{L}_\pi\|$). *Suppose Assumption 3.5 holds. Then for any $\boldsymbol{\omega} \in \mathbb{R}^d$, we have*

$$\|\nabla\mathcal{L}(\boldsymbol{\omega})\|^2 \leq 2 \|\nabla\mathcal{L}_\pi(\boldsymbol{\omega})\|^2 + \mathcal{O}(d\epsilon^2). \quad (24)$$

The proof is provided in Appendix F. This result bridges the smoothed objective $\mathcal{L}_\pi$ and the original objective $\mathcal{L}$, allowing convergence guarantees proved for $\mathcal{L}_\pi$ to be carried over to $\mathcal{L}$. We formalize this transfer in the following theorem.

**Theorem 3.8.** *Let $\{\boldsymbol{\omega}_t\}_{t\geq 0}$ be the sequence of iterates generated by the proposed CurvZO with stepsize $\eta = \frac{1}{3L}$ and variance-minimizing sampling rule $\{\pi_i^t\}_{i=1}^d$ satisfying $\sum_{i=1}^d \pi_i^t = B$. Then we have:*

$$\min_{0 \leq t \leq T} \mathbb{E}\left[\|\nabla\mathcal{L}(\boldsymbol{\omega}_t)\|^2\right] \leq \frac{12L\left(\mathcal{L}(\boldsymbol{\omega}_0) - \mathcal{L}^*\right)}{T+1} + \frac{2CM^2}{3B} + \mathcal{O}(d\epsilon^2), \quad (25)$$

*where $M$ is an upper bound on $\sum_{i=1}^d \sqrt{F_{ii}(\boldsymbol{\omega}_t)}$, and $C > 0$ is a constant arising from the variance bound in Proposition 3.3.*

This theorem establishes that the curvature-guided sparse zeroth-order method achieves an $\mathcal{O}(1/T)$ convergence rate for the expected squared gradient norm, up to a variance floor of order $\mathcal{O}(M^2/B)$ and a smoothing bias term of order $\mathcal{O}(d\epsilon^2)$. The proof is provided in Appendix G.

### 3.5. Block-Wise Curvature Scores

Tracking curvature scores at the per-parameter level incurs substantial computational and memory overhead. To mitigate this, we group parameters into disjoint blocks (e.g., tensors or layers) and adopt a block-wise formulation that mirrors the per-parameter approach.

We partition the parameters into $G$ disjoint blocks $\{\mathcal{G}_1, \ldots, \mathcal{G}_G\}$. Let $\boldsymbol{m}^{\text{blk}} \in \{0,1\}^G$ with independent entries $m_i^{\text{blk}} \sim \text{Bernoulli}(\pi_i^{\text{blk}})$. Given $\boldsymbol{z} \sim \mathcal{N}(\boldsymbol{0}, \boldsymbol{I}_d)$, we define the block-wise sparse perturbation $\boldsymbol{v}^{\text{blk}} \in \mathbb{R}^d$ by

$$\boldsymbol{v}_{\mathcal{G}_i}^{\text{blk}} \triangleq m_i^{\text{blk}} \boldsymbol{z}_{\mathcal{G}_i}, \quad i = 1, \ldots, G, \quad (26)$$

where $\boldsymbol{z}_{\mathcal{G}_i} \triangleq \text{concat}\{z_j : j \in \mathcal{G}_i\} \in \mathbb{R}^{|\mathcal{G}_i|}$.

We define the block-wise curvature score for block $\mathcal{G}_i$ as:

$$\tilde{s}_i^{\text{blk}} \triangleq \frac{\|\boldsymbol{v}_{\mathcal{G}_i}^{\text{blk}}\|_2^2}{\|\boldsymbol{v}^{\text{blk}}\|_2^2} \Delta^2. \quad (27)$$

We smooth $\{\tilde{s}_i^{\text{blk}}\}_{i=1}^G$ over iterations using EMA to obtain the smoothed block-level curvature scores $\{S_i^{\text{blk},t}\}_{i=1}^G$, as in Eq. (8).

Given the response $\Delta$, the block-wise gradient estimator is defined as

$$\tilde{\boldsymbol{g}}_{\mathcal{G}_i} = \frac{\Delta}{\pi_i^{\text{blk}}} m_i^{\text{blk}} \boldsymbol{z}_{\mathcal{G}_i}. \quad i = 1, \ldots, G, \quad (28)$$

**Proposition 3.9** (Block-wise Consistency). *For each parameter block $\mathcal{G}_i$, the block-wise gradient estimator $\tilde{\boldsymbol{g}}_{\mathcal{G}_i}$ in Eq. (28) satisfies*

$$\mathbb{E}_{\boldsymbol{m}^{\text{blk}},\boldsymbol{z}}[\tilde{\boldsymbol{g}}_{\mathcal{G}_i}] = \boldsymbol{g}_{\mathcal{G}_i} + \mathcal{O}(\epsilon^2), i = 1, \ldots, G, \quad (29)$$

*Table 1.* Accuracy (%) on OPT-2.7B fine-tuning with 1,000 training samples. FT and LoRA are first-order (backpropagation-based) oracle baselines. Better results among MeZO, DiZO, and CurvZO are highlighted in **bold**.

| Task | SST-2 | RTE | CB | BoolQ | WSC | WIC | SQuAD | DROP | Average |
|------|-------|-----|-----|-------|-----|-----|-------|------|---------|
| | | | -classification- | | | | -generation- | | |
| Zero-shot | 56.3 | 54.1 | 50.0 | 47.6 | 36.5 | 52.6 | 29.8 | 9.5 | 42.1 |
| FT | 93.8 | 80.1 | 82.1 | 72.6 | 63.4 | 69.1 | 82.9 | 33.2 | 72.2 |
| LoRA | 94.7 | 81.6 | 82.7 | 74.1 | 59.6 | 62.1 | 84.5 | 35.1 | 71.8 |
| MeZO | 92.1 | 63.3 | 67.8 | 66.7 | 58.6 | 59.8 | 79.0 | 26.5 | 64.2 |
| DiZO | **94.4** | 65.6 | 66.0 | 66.8 | 56.7 | 57.1 | 62.1 | 22.5 | 61.4 |
| **CurvZO** | 94.1 | **67.7** | **69.8** | **68.9** | **62.9** | **61.9** | **81.5** | **27.6** | **66.8** |
| MeZO + LoRA | 93.0 | 58.1 | 64.2 | 63.2 | 56.7 | 53.7 | 80.8 | 26.9 | 62.1 |
| DiZO + LoRA | 92.0 | 55.5 | 64.2 | 64.0 | 58.6 | 54.2 | 67.5 | 22.3 | 59.8 |
| **CurvZO + LoRA** | **93.6** | **61.7** | **69.6** | **66.1** | **63.4** | **57.6** | **81.1** | **27.6** | **65.1** |

*Table 2.* Accuracy (%) on OPT-6.7B fine-tuning with 1,000 training samples.

| Task | SST-2 | RTE | WIC | WSC | SQuAD |
|------|-------|-----|-----|-----|-------|
| | | -classification- | | | -generation- |
| MeZO | 94.1 | 70.3 | **62.5** | 60.5 | 81.6 |
| DiZO | 91.6 | 66.4 | 61.6 | 59.6 | 67.8 |
| **CurvZO** | **94.7** | **72.2** | 61.7 | **61.5** | **83.7** |
| MeZO + LoRA | 93.0 | **64.9** | 61.4 | 60.5 | 80.1 |
| DiZO + LoRA | 91.4 | 58.8 | 57.2 | 58.6 | 71.5 |
| **CurvZO + LoRA** | 93.1 | 63.5 | **61.5** | 63.4 | 80.6 |

where $\boldsymbol{g}_{\mathcal{G}_i} = \operatorname{concat}\{g_j : j \in \mathcal{G}_i\} \in \mathbb{R}^{|\mathcal{G}_i|}$.

*Moreover, let $F_i^{\mathrm{blk}} = \sum_{j \in \mathcal{G}_i} F_{jj}$ denote the block-wise diagonal Fisher quantity. The block-level variance of $\tilde{\boldsymbol{g}}_{\mathcal{G}_i} \triangleq \sum_{j \in \mathcal{G}_i} \operatorname{Var}(\tilde{g}_j)$ admits the upper bound*

$$\operatorname{Var}(\tilde{\boldsymbol{g}}_{\mathcal{G}_i}) \leq C \frac{F_i^{\mathrm{blk}}}{\pi_i^{\mathrm{blk}}}, \tag{30}$$

*for some constant $C > 0$.*

Analogous to Problem (P1), we construct the block sampling probabilities $\boldsymbol{\pi}^{\mathrm{blk}}$ by solving the optimization problem (P2), which minimizes the total variance upper bound $\sum_{i=1}^{G} \operatorname{Var}(\tilde{\boldsymbol{g}}_{\mathcal{G}_i})$ subject to the budget constraint $B = \sum_{i=1}^{G} \pi_i^{\mathrm{blk}}$.

**Proposition 3.10** (Block-wise Variance-Minimizing Sampling Distribution). *Let $\boldsymbol{\pi}^{\mathrm{blk},\star} = \{\pi_i^{\mathrm{blk}\star}\}_{i=1}^{G}$ be an optimal solution to Problem (P2), it satisfies*

$$\pi_i^{\mathrm{blk}\star} \propto \sqrt{F_i^{\mathrm{blk}}}, \qquad i = 1, \dots, G, \tag{31}$$

*which is the block-level counterpart of the per-parameter rule $\pi_j^{\star} \propto \sqrt{F_{jj}}$, obtained by aggregating diagonal Fisher information within each block as $F_i^{\mathrm{blk}} = \sum_{j \in \mathcal{G}_i} F_{jj}$.*

Using block-wise curvature scores preserves the same sampling rule. Specifically, we replace per-parameter scores with block-level scores and compute block sampling probabilities via the variance-motivated rule: $\pi_i^{\mathrm{blk}} \propto \sqrt{S_i^{\mathrm{blk}}}$.

Moreover, the budget selection depends only on the normalized curvature score distribution induced by $\{S_i^{\mathrm{blk}}\}_{i=1}^{G}$, not on within-block dimensionality. Therefore, the same adaptivity indicators extend naturally to the block level by replacing the total parameter count $d$ with the number of blocks $G$: the effective support $d_{\mathrm{eff}}$ now measures the effective number of blocks that contribute to the curvature score distribution, while the entropy $H$ captures its sharpness. The proofs of Propositions 3.9 and 3.10 are provided in Appendix H.

## 4. Experiments

### 4.1. Experimental Settings

We experiment with two open-source LLM families of different sizes (Zhang et al., 2022; Touvron et al., 2023), including OPT (2.7B, 6.7B) and Llama2 (7B, 13B), and evaluate them mainly on SuperGLUE (Wang et al., 2019). We primarily compare against two representative ZO baselines, MeZO (Malladi et al., 2023) and DiZO (Tan et al., 2026), and also consider parameter-efficient fine-tuning via LoRA (Hu et al., 2022). All experiments use block-wise curvature scores, where each block corresponds to one of the model's native parameter tensors. Experimental details are provided in Appendix I. To further assess the generality of CurvZO, Appendix J includes additional experiments on Llama3.1-8B, more challenging benchmarks such as Dolly-15k (Conover et al., 2023) and GSM8K (Cobbe et al., 2021), additional ZO baselines such as Sparse-MeZO (Liu et al., 2026) and SensZOQ (Guo et al., 2025), and larger training sets. Further ablation studies are provided in Appendix K. Our code is released at https://github.com/Eurekabb/CurvZO.git.

### 4.2. Performance

We evaluate the generalizability of CurvZO across different model architectures and scales. Fine-tuning accuracy results are reported for OPT-2.7B and OPT-6.7B in Tables 1 and 2, respectively, and for the Llama series in Figure 2. In addition, we compare the convergence behavior, memory usage,

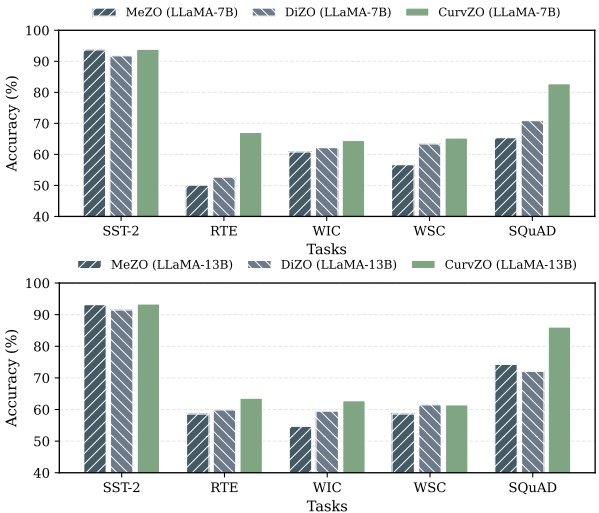

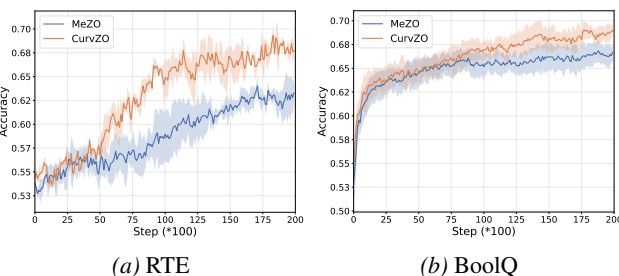

*Figure 2.* Accuracy (%) on fine-tuning Llama2-7B (top) and Llama2-13B (bottom) with 1,000 training samples.

*(a)* RTE        *(b)* BoolQ

*Figure 3.* Convergence curves of fine-tuning OPT-2.7B with MeZO and CurvZO on (a) RTE, (b) BoolQ tasks.

and GPU hours of CurvZO and MeZO on OPT-2.7B across tasks: convergence curves are shown in Figure 3, memory usage is reported in Table 3, and GPU-hour comparisons are provided in Figure 4.

We highlight the key experimental findings as follows.

**CurvZO outperforms baselines in both standard and parameter-efficient settings.** As detailed in Table 1, CurvZO consistently surpasses MeZO and DiZO, with or without LoRA, while achieving performance competitive with FO methods. Specifically, it achieves the best results on five out of six classification benchmarks and leads in both text generation tasks. For instance, CurvZO achieves notable improvements on WSC (↑ 4.3%), SQuAD (↑ 2.5%) and BoolQ (↑ 2.1%) demonstrating its robustness across diverse tasks. Similarly, in the LoRA setting, CurvZO yields substantial gains on CB (↑ 5.4%), WSC (↑ 4.8%), and RTE (↑ 3.6%). Crucially, these performance gains extend to larger scales (OPT-6.7B, Table 2) and different architectures, as evidenced by consistent improvements on both Llama2-7B and Llama2-13B (Figure 2).

**CurvZO converges in fewer iterations and achieves lower training GPU hours than MeZO.** As shown in Figure 3,

*Table 3.* Memory usage (batch size = 1) of fine-tuning OPT-2.7B and OPT-6.7B (1,000 examples). For OPT-6.7B, FT exceeds the 80 GB memory limit of a single GPU, while MeZO and CurvZO remain memory-efficient.

| Model | Task | SST-2 | RTE | WIC | WSC | SQuAD |
|-------|------|-------|-----|-----|-----|-------|
| | | | —classification— | | | -generation- |
| OPT-2.7B | FT | 42.88 | 45.28 | 43.48 | 43.21 | 45.28 |
| OPT-2.7B | MeZO | 5.91 | 5.91 | 5.91 | 5.91 | 5.91 |
| OPT-2.7B | CurvZO | 5.91 | 5.91 | 5.91 | 5.91 | 5.91 |
| OPT-6.7B | FT | > 80.00 | > 80.00 | > 80.00 | > 80.00 | > 80.00 |
| OPT-6.7B | MeZO | 13.94 | 13.94 | 13.94 | 13.94 | 13.94 |
| OPT-6.7B | CurvZO | 13.95 | 13.95 | 13.95 | 13.95 | 13.95 |

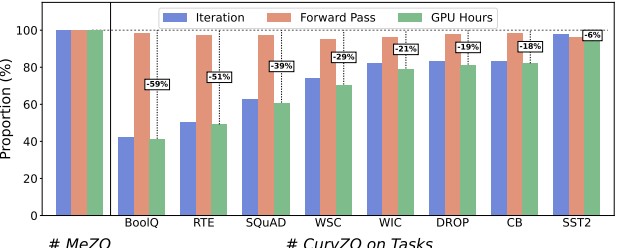

*Figure 4.* GPU hours of MeZO and CurvZO across tasks, together with convergence iterations and forward pass time.

CurvZO reaches the same accuracy as MeZO in approximately $2.4\times$ fewer optimization steps on BoolQ and $2.0\times$ fewer steps on RTE. Crucially, this faster convergence translates directly into reduced computational cost: Figure 4 shows that CurvZO consistently reduces total GPU hours across all tasks, with savings of up to 59% and 51% on BoolQ and RTE, respectively, demonstrating substantial efficiency gains.

**Memory efficiency on par with MeZO.** Table 3 shows that CurvZO uses essentially the same GPU memory as MeZO: identical on OPT-2.7B and only slightly higher on OPT-6.7B due to storing curvature scores. This overhead is negligible compared to full fine-tuning, while CurvZO delivers better accuracy and faster convergence than MeZO.

## 5. Conclusion

With LLMs rapidly scaling, memory-efficient fine-tuning is increasingly important. ZO optimization avoids backpropagation but often suffers from slow or unstable convergence due to high-variance gradient estimates. Sparse perturbations help, yet their effectiveness depends critically on which parameters are selected for perturbation under a limited budget. We propose **CurvZO**, which tracks online curvature scores from scalar ZO feedback to select parameters for sparse perturbations, and further adapts the perturbation budget based on the evolving score distribution. Extensive experiments on OPT and Llama across diverse NLP tasks show that CurvZO consistently outperforms ZO baselines and converges faster, improving accuracy by up to 4.4 points and reducing GPU hours by up to $2\times$, while maintaining ZO-level memory efficiency.

# 6. Limitations

We stabilize the curvature score to reduce perturbation-scale noise and temporal variability, and introduce adaptive budget selection to balance exploration and exploitation based on the evolving scores. While these designs improve the reliability of curvature tracking and lead to more stable and efficient sparse updates, our curvature score is computed per coordinate/block and reflects only local sensitivity. As a result, it does not capture coupling between parameters. This limitation may result in mildly suboptimal sampling in some regimes where interaction-aware second-order information is important, potentially leaving some performance gains unrealized.

# Acknowledgments

This work was supported in part by the National Natural Science Foundation of China under Grant 62202214 and the Guangdong Basic and Applied Basic Research Foundation under Grant 2023A1515012819.

# Impact Statement

This work introduces CurvZO, which tracks curvature signals to guide sparse ZO updates for memory-efficient LLM fine-tuning. By improving the efficiency and scalability of ZO fine-tuning, it may broaden access to model adaptation on resource-constrained hardware and reduce the cost of practical deployment. CurvZO is an optimization method and does not introduce direct societal risks on its own.

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

## A. Proof of Eq. (6)

The proof expands $\Delta^2 \approx (\boldsymbol{g}^T \boldsymbol{v})^2$ and exploits the independence of the perturbation components $\boldsymbol{v}$, i.e., $z_i$ and $m_j$ are independent with $z_i \sim \mathcal{N}(0,1)$ and $m_j \sim \text{Bernoulli}(\pi_j)$. Cross-terms vanish in expectation due to $\mathbb{E}[v_i] = 0$, while the diagonal terms are evaluated using the moments $\mathbb{E}[v_i^4] = 3\pi_i$ and $\mathbb{E}[v_j^2 v_i^2] = \pi_j \pi_i$, establishing the identity.

Given the expansion $\Delta = \sum_{j=1}^d g_j v_j + \mathcal{O}(\epsilon^2)$, we compute the expectation of the curvature score $s_i = \Delta^2 v_i^2$ with respect to the perturbation $\boldsymbol{v}$:

$$
\begin{aligned}
&\mathbb{E}_v[\Delta^2 v_i^2] = \\
&\mathbb{E}_v\left[\left(\sum_{j=1}^d g_j v_j\right)\left(\sum_{k=1}^d g_k v_k\right) v_i^2\right] + \mathcal{O}(\epsilon^2) \\
&= \sum_{j=1}^d \sum_{k=1}^d g_j g_k \mathbb{E}[v_j v_k v_i^2] + \mathcal{O}(\epsilon^2).
\end{aligned}
\tag{32}
$$

We then evaluate the expectation term $E_{j,k,i} = \mathbb{E}[v_j v_k v_i^2]$.

**Case 1:** $j \neq k$. Since the components of $\boldsymbol{v}$ are independent and $\mathbb{E}[z_i] = 0$ (owing to $z_i \sim \mathcal{N}(0,1)$ and the independent of $m_i$ and $z_i$), we have

$$
\mathbb{E}[v_l] = \mathbb{E}[m_l \cdot z_l] = \mathbb{E}[m_l] \cdot \mathbb{E}[z_l] = 0. \tag{33}
$$

Due to the mutual independence of distinct coordinates, for any $j \neq k$, we have

$$
\begin{aligned}
\mathbb{E}[v_j v_k v_i^2] &= \mathbb{E}[v_j] \cdot \mathbb{E}[v_k] \cdot \mathbb{E}[v_i^2] \\
&= 0 \cdot 0 \cdot \mathbb{E}[v_i^2] \\
&= 0, \quad \forall j \neq k.
\end{aligned}
\tag{34}
$$

**Case 2:** $j = k$. The expression simplifies as

$$
\sum_{j=1}^d g_j^2 \mathbb{E}[v_j^2 v_i^2]. \tag{35}
$$

We assume that $m$ and $z$ are independent, with $z_i \sim \mathcal{N}(0,1)$ and $m_i \sim \text{Bernoulli}(\pi_i)$. Under this assumption, we have

$$
\mathbb{E}[z_i^2] = 1, \quad \mathbb{E}[z_i^4] = 3. \tag{36}
$$

*Subcase 2a:* $j \neq i$.

$$
\begin{aligned}
\mathbb{E}[v_j^2 v_i^2] &= \mathbb{E}[v_j^2]\mathbb{E}[v_i^2] \\
&= \mathbb{E}[m_j^2]\mathbb{E}[m_i^2]\mathbb{E}[z_j^2]\mathbb{E}[z_i^2] \\
&= \pi_j \cdot \pi_i.
\end{aligned}
\tag{37}
$$

*Subcase 2b:* $j = i$. Since $m_i \in \{0,1\}$, we have $m_i^4 = m_i$ and

$$
\begin{aligned}
\mathbb{E}[v_i^4] &= \mathbb{E}[m_i^4]\mathbb{E}[z_i^4] \\
&= \mathbb{E}[m_i]\mathbb{E}[z_i^4] \\
&= 3 \cdot \pi_i.
\end{aligned}
\tag{38}
$$

Substituting the results from Subcases 2a and 2b back into $\mathbb{E}_v[\Delta^2 v_i^2]$,

$$
\begin{aligned}
\sum_{j=1}^d g_j^2 \mathbb{E}[v_j^2 v_i^2] &= \sum_{j \neq i} g_j^2(\pi_j \pi_i) + g_i^2(3\pi_i) \\
&= \pi_i \left(\sum_{j \neq i} \pi_j g_j^2\right) + 3\pi_i g_i^2.
\end{aligned}
\tag{39}
$$

To recover the full summation form $\sum_{j=1}^d \pi_j g_j^2$, we add and subtract the term for $j = i$ (which is $\pi_i^2 g_i^2$) inside the parenthesis:

$$
\begin{aligned}
\mathbb{E}_{\boldsymbol{v}}[\Delta^2 v_i^2] &= \pi_i \left(\sum_{j \neq i} \pi_j g_j^2 + \pi_i g_i^2\right) \\
&\quad - \pi_i(\pi_i g_i^2) + 3\pi_i g_i^2 + \mathcal{O}(\epsilon^2) \\
&= \pi_i \left(\sum_{j=1}^d \pi_j g_j^2\right) + (3\pi_i - \pi_i^2)g_i^2 + \mathcal{O}(\epsilon^2) \\
&= \pi_i \sum_{j=1}^d \pi_j g_j^2 + \pi_i(3 - \pi_i)g_i^2 + \mathcal{O}(\epsilon^2).
\end{aligned}
\tag{40}
$$

## B. Variance Analysis of the Curvature Score and its Unbiased Variant

We define an unbiased variant of curvature score $s_i$ as:

$$
s_i' = \frac{s_i - \pi_i \Delta^2}{\pi_i(3 - \pi_i)}. \tag{41}
$$

To show that $s_i'$ is unbiased, we take expectation over the perturbation randomness (i.e., $\boldsymbol{m}$ and $\boldsymbol{z}$), obtaining

$$
\mathbb{E}_{\boldsymbol{m},\boldsymbol{z}}[s_i'] = g_i^2 + \mathcal{O}(\epsilon^2). \tag{42}
$$

Then, we analyze the variance of the curvature score $s_i$ and its unbiased variant $s_i'$, yielding the following results:

- **Stability of the $s_i$:** $\text{Var}(s_i)$ vanishes as $\pi_i \to 0$; in particular, $\text{Var}(s_i) = \mathcal{O}(\pi_i)$.

- **Instability of $s_i'$:** $\text{Var}(s_i')$ diverges as $\pi_i \to 0$; in particular, $\text{Var}(s_i') = \mathcal{O}(1/\pi_i)$.

Consequently, $s_i$ is better suited than $s_i'$ for tracking curvature signals in LLMs. In LLMs, the parameter dimension is enormous and local curvature magnitudes are highly heterogeneous, so a curvature-guided sampling distribution must allocate most of the sampling probability to a tiny subset of high-curvature parameters, leaving the vast majority with extremely small sampling probabilities. In this regime, the low-variance behavior of $s_i$ is essential, whereas the variance of $s_i'$ becomes prohibitively large.

We provide detailed derivations of the variances of the curvature score $s_i$ and its unbiased variant $s_i'$ in the following subsections.

### B.1. Variance of the Curvature Score $s_i$

We analyze $\text{Var}(s_i)$ via $\text{Var}(s_i) = \mathbb{E}[s_i^2] - (\mathbb{E}[s_i])^2$:

$$\mathbb{E}[s_i^2] = \mathbb{E}\left[(\Delta^2 v_i^2)^2\right] = \mathbb{E}\left[\Delta^4 v_i^4\right]$$
$$= \mathbb{E}\left[\left(\sum_{k=1}^{d} g_k v_k\right)^4 v_i^4\right]. \tag{43}$$

We isolate the $k = i$ term from the sum, i.e., $\sum_{k=1}^{d} g_k v_k = g_i v_i + \sum_{k \neq i} g_k v_k$. Then, we have

$$\mathbb{E}[s_i^2] = C \sum_{k=0}^{4} \binom{4}{k} g_i^k \mathbb{E}\left[v_i^{4+k}\left(\sum_{j \neq i} g_j v_j\right)^{4-k}\right]$$
$$= C \sum_{k=0}^{4} \binom{4}{k} g_i^k \mathbb{E}[v_i^{4+k}] \cdot \mathbb{E}\left[\left(\sum_{j \neq i} g_j v_j\right)^{4-k}\right], \tag{44}$$

where $C$ is a constant.

Recall that $v_i = m_i z_i$, where $m_i \sim \text{Bernoulli}(\pi_i)$ and $z_i \sim \mathcal{N}(0, 1)$. The moment of $v_i$ is given by:

$$\mathbb{E}[v_i^p] = \mathbb{E}[m_i^p]\mathbb{E}[z_i^p] = \pi_i \mathbb{E}[z_i^p]. \tag{45}$$

We note that the expectation of the cross terms with $j \neq i$ can be bounded by a constant $M$ since they involve only finite moments of bounded (or moment-bounded) random variables. Therefore, we have

$$\mathbb{E}[s_i^2] = \sum_{k=0}^{4} C_k (\pi_i \mathbb{E}[z_i^{4+k}]) \underbrace{\mathbb{E}\left[\left(\sum_{j \neq i} g_j v_j\right)^{4-k}\right]}_{\text{bounded by } M} \tag{46}$$

$$= \pi_i \cdot \mathcal{O}(1) = \mathcal{O}(\pi_i).$$

Since $\text{Var}(s_i) \leq \mathbb{E}[s_i^2]$, we obtain

$$\text{Var}(s_i) = \mathcal{O}(\pi_i). \tag{47}$$

In particular, $\text{Var}(s_i) \to 0$ as $\pi_i \to 0$.

### B.2. Variance of Unbiased Variant $s_i'$

Let $Y_i \triangleq \Delta^2(v_i^2 - \pi_i)$. Then, the variance of $Y_i$ is given by:

$$\text{Var}(s_i') = \frac{1}{\pi_i^2(3 - \pi_i)^2}\text{Var}(Y_i). \tag{48}$$

The second moment of $Y_i$ is

$$\mathbb{E}[Y_i^2] = \mathbb{E}\left[\Delta^4(v_i^2 - \pi_i)^2\right]$$
$$= \pi_i \mathbb{E}[Y_i^2 \mid m_i = 1] + (1 - \pi_i)\mathbb{E}[Y_i^2 \mid m_i = 0]. \tag{49}$$

**Case (a):** $m_i = 1$. In this case, $v_i = z_i$ $(v_i^2 - \pi_i)^2 = (z_i^2 - \pi_i)^2$.

$$\mathbb{E}[Y_i^2 \mid m_i = 1] =$$
$$\mathbb{E}\left[\left(g_i z_i + \sum_{j \neq i} g_j v_j\right)^4 (z_i^2 - \pi_i)^2\right] \leq M_1. \tag{50}$$

Since all moments of $z_i$ and $v_j$ are finite, this term is bounded by some constant $M_1$.

**Case (b):** $m_i = 0$. In this case, $(v_i^2 - \pi_i)^2 = (0 - \pi_i)^2 = \pi_i^2$.

$$\mathbb{E}[Y_i^2 \mid m_i = 0] = \mathbb{E}\left[\left(\sum_{j \neq i} g_j v_j\right)^4 (\pi_i^2)\right]$$
$$= \pi_i^2 \cdot \mathbb{E}\left[\left(\sum_{j \neq i} g_j v_j\right)^4\right] \leq M_2. \tag{51}$$

The expectation term is bounded by some constant $M_2$. Substituting back into the expression for $\mathbb{E}[Y_i^2]$, we have

$$\mathbb{E}[Y_i^2] = \pi_i M_1 + (1 - \pi_i)\pi_i^2 M_2 \approx \pi_i M_1. \tag{52}$$

$$\text{Var}(s_i') \approx \frac{\mathbb{E}[Y_i^2]}{\pi_i^2(3 - \pi_i)^2} \approx \frac{\pi_i M_1 + \mathcal{O}(\pi_i^2)}{\pi_i^2 \cdot 9}$$
$$= \mathcal{O}\left(\frac{1}{\pi_i}\right). \tag{53}$$

Therefore, as $\pi_i \to 0$, $\text{Var}(s_i') \to \infty$.

## C. Proof of Proposition 3.2

We prove unbiasedness by Taylor expanding the loss to approximate the finite-difference response $\Delta$. Using the independence across perturbation coordinates and the Horvitz–Thompson correction, we show that the cross terms vanish in expectation, while the diagonal terms recover the true gradient up to an $\mathcal{O}(\epsilon^2)$ bias.

Under the assumption that $\mathcal{L}$ is sufficiently smooth, a Taylor expansion gives:

$$\mathcal{L}(\boldsymbol{\omega} + \epsilon\boldsymbol{v}) = \mathcal{L}(\boldsymbol{\omega}) + \epsilon\nabla\mathcal{L}(\boldsymbol{\omega})^T\boldsymbol{v}$$
$$+ \frac{\epsilon^2}{2}\boldsymbol{v}^T\nabla^2\mathcal{L}(\boldsymbol{\omega})\boldsymbol{v} + \mathcal{O}(\epsilon^3) \quad (54)$$

$$\mathcal{L}(\boldsymbol{\omega} - \epsilon\boldsymbol{v}) = \mathcal{L}(\boldsymbol{\omega}) - \epsilon\nabla\mathcal{L}(\boldsymbol{\omega})^T\boldsymbol{v}$$
$$+ \frac{\epsilon^2}{2}\boldsymbol{v}^T\nabla^2\mathcal{L}(\boldsymbol{\omega})\boldsymbol{v} - \mathcal{O}(\epsilon^3) \quad (55)$$

Since $\boldsymbol{g} = \nabla\mathcal{L}(\boldsymbol{\omega})$, subtracting the two equations yields:

$$\mathcal{L}(\boldsymbol{\omega} + \epsilon\boldsymbol{v}) - \mathcal{L}(\boldsymbol{\omega} - \epsilon\boldsymbol{v}) = 2\epsilon\boldsymbol{g}^T\boldsymbol{v} + \mathcal{O}(\epsilon^3). \quad (56)$$

Thus,

$$\Delta = \frac{\mathcal{L}(\boldsymbol{\omega} + \epsilon\boldsymbol{v}) - \mathcal{L}(\boldsymbol{\omega} - \epsilon\boldsymbol{v})}{2\epsilon} = \boldsymbol{g}^T\boldsymbol{v} + \mathcal{O}(\epsilon^2). \quad (57)$$

Recall that $\Delta = \sum_{j=1}^{d} g_j v_j + \mathcal{O}(\epsilon^2)$. Substitute this expression into the definition of the estimator $\tilde{g}_i$:

$$\tilde{g}_i = \left(\sum_{j=1}^{d} g_j v_j + \mathcal{O}(\epsilon^2)\right)\frac{v_i}{\pi_i}, \quad i = 1,\dots,d. \quad (58)$$

Taking expectation over the perturbation randomness yields:

$$\mathbb{E}[\tilde{g}_i] = \frac{1}{\pi_i}\sum_{j=1}^{d} g_j\mathbb{E}[v_j v_i] + \frac{1}{\pi_i}\mathbb{E}[v_i \cdot \mathcal{O}(\epsilon^2)]. \quad (59)$$

Since $v_k = m_k z_k$, $\mathbb{E}[\tilde{g}_i]$ becomes

$$\mathbb{E}[\tilde{g}_i] = \frac{1}{\pi_i}\sum_{j=1}^{d} g_j\mathbb{E}[m_j m_i]\mathbb{E}[z_j z_i] + \frac{1}{\pi_i}\mathbb{E}[v_i \cdot \mathcal{O}(\epsilon^2)]$$
$$= \frac{1}{\pi_i}\sum_{j=1}^{d} g_j\mathbb{E}[m_j m_i]\delta_{ij} + \mathcal{O}(\epsilon^2). \quad (60)$$

Using the property that terms with $j \neq i$ vanish (since $\delta_{ij} = 0$ for $j \neq i$), we have

$$\mathbb{E}[\tilde{g}_i] = \frac{1}{\pi_i}g_i\mathbb{E}[m_i^2] \cdot 1 + \mathcal{O}(\epsilon^2)$$
$$= \frac{1}{\pi_i}g_i\pi_i + \mathcal{O}(\epsilon^2) \quad (61)$$
$$= g_i + \mathcal{O}(\epsilon^2).$$

Finally, aggregating over all coordinates gives $\mathbb{E}[\tilde{\boldsymbol{g}}] = \boldsymbol{g} + \mathcal{O}(\epsilon^2)$.

## D. Proof of Proposition 3.3

By definition of unbiased gradient estimator $\tilde{g}_i = (\Delta v_i)/\pi_i$, we have $\mathbb{E}[\tilde{g}_i^2] = \mathbb{E}[\Delta^2 v_i^2]/\pi_i^2$. Substituting the Taylor

expansion $\Delta = \boldsymbol{g}^\top\boldsymbol{v} + \mathcal{O}(\epsilon^2)$ yields $\mathbb{E}[\tilde{g}_i^2] = \mathcal{O}(F_{ii}/\pi_i)$ where $F_{ii}$ denotes the diagonal Fisher information matrix.

Since $\tilde{g}_i = (\Delta m_i z_i)/\pi_i$ and $v_i = m_i z_i$, we have

$$\mathbb{E}[\tilde{g}_i^2] = \frac{1}{\pi_i^2}\mathbb{E}[\Delta^2 v_i^2]. \quad (62)$$

As shown in Eq. (6), we have

$$\mathbb{E}_{\boldsymbol{v}}[\Delta^2 v_i^2] = 3\pi_i g_i^2 + \pi_i\sum_{j\neq i}\pi_j g_j^2, \quad (63)$$

where we omit the $\mathcal{O}(\epsilon^2)$ term in Eq. (6) for simplicity. Therefore,

$$\mathbb{E}[\tilde{g}_i^2] = \frac{1}{\pi_i}\left(3g_i^2 + \sum_{j\neq i}\pi_j g_j^2\right). \quad (64)$$

Taking expectations and denoting $F_{ii} = \mathbb{E}[g_i^2]$ yields

$$\text{Var}(\tilde{g}_i) \leq \frac{3}{\pi_i}F_{ii} + \frac{1}{\pi_i}\sum_{j\neq i}\pi_j F_{jj} \leq \frac{3F_{ii} + \sum_j \pi_j F_{jj}}{\pi_i}. \quad (65)$$

The bound in (65) consists of two parts. The first term, $3F_{ii}/\pi_i$, captures the Fisher-dependent contribution of coordinate $i$. The second term, $\sum_j \pi_j F_{jj}/\pi_i$, depends on a global Fisher-weighted aggregate shared across coordinates. Therefore, for our purpose of deriving a coordinate-adaptive sampling rule, we focus on the coordinate-wise term and absorb the shared term into a constant:

$$\text{Var}(\tilde{g}_i) \leq \frac{C_0 + C_1 F_{ii}}{\pi_i}, \quad (66)$$

where $C_1 = 3$ and $C_0 \approx \sum_j \pi_j F_{jj}$ is (up to a negligible dependence on $i$) independent of the index $i$. This approximation is motivated by two considerations. First, from a sampling-design perspective, the coordinate-wise term determines how sampling probabilities should vary across coordinates, which is exactly what CurvZO needs to allocate a limited perturbation budget via the sampling rule. Second, retaining the shared term leads to a coupled optimization problem without a simple closed-form solution, making the rule impractical for online use. Since ZO fine-tuning uses noisy online curvature scores rather than exact Fisher diagonals, we adopt the simplified form to preserve the main coordinate-wise dependence while keeping the rule tractable.

Because the sampling probabilities $\pi_i$ are design variables under our control for all coordinates $i$, and the global term $C_0$ affects all coordinates in a similar way, the decomposition in (66) implies the following variance scaling:

$$\text{Var}(\tilde{g}_i) = \mathcal{O}\left(\frac{F_{ii}}{\pi_i}\right), \quad (67)$$

up to a coordinate-independent multiplicative constant.

# E. Proof of Proposition 3.4

We solve Problem (P1) via a Lagrangian approach, which yields the optimal sampling rule $\pi_i^\star \propto \sqrt{F_{ii}}$.

Introducing the Lagrangian

$$\mathcal{J}(\boldsymbol{\pi}, \lambda) = \sum_{i=1}^{d} \frac{F_{ii}}{\pi_i} + \lambda \left( \sum_{i=1}^{d} \pi_i - B \right), \qquad (68)$$

and setting the derivative with respect to each $\pi_i$ to zero yields

$$\frac{\partial \mathcal{L}}{\partial \pi_i} = -\frac{F_{ii}}{\pi_i^2} + \lambda = 0 \implies \pi_i^2 = \frac{F_{ii}}{\lambda}. \qquad (69)$$

Therefore, $\pi_i^\star \propto \sqrt{F_{ii}}$.

# F. Proof of Lemma 3.7.

Using the norm inequality induced by Young's inequality, $\|\boldsymbol{a} + \boldsymbol{b}\|^2 \leq 2\|\boldsymbol{a}\|^2 + 2\|\boldsymbol{b}\|^2$, we obtain

$$\|\nabla \mathcal{L}(\boldsymbol{\omega})\|^2 \leq 2\|\nabla \mathcal{L}_\pi(\boldsymbol{\omega})\|^2 + 2\|\nabla \mathcal{L}(\boldsymbol{\omega}) - \nabla \mathcal{L}_\pi(\boldsymbol{\omega})\|^2. \qquad (70)$$

Let Bias $= \|\nabla \mathcal{L}(\boldsymbol{\omega}) - \nabla \mathcal{L}_\pi(\boldsymbol{\omega})\|$. Recall $\nabla \mathcal{L}_\pi(\boldsymbol{\omega}) = \mathbb{E}_{\boldsymbol{m},\boldsymbol{z}}[\nabla \mathcal{L}(\boldsymbol{\omega} + \epsilon \boldsymbol{v})]$. By Jensen's inequality and Lipschitz continuity:

$$\begin{aligned} \text{Bias} &= \|\mathbb{E}_{\boldsymbol{m},\boldsymbol{z}}[\nabla \mathcal{L}(\boldsymbol{\omega}) - \nabla \mathcal{L}(\boldsymbol{\omega} + \epsilon \boldsymbol{v})]\| \\ &\leq \mathbb{E}_{\boldsymbol{m},\boldsymbol{z}}[\|\nabla \mathcal{L}(\boldsymbol{\omega}) - \nabla \mathcal{L}(\boldsymbol{\omega} + \epsilon \boldsymbol{v})\|] \qquad (71) \\ &\leq \mathbb{E}_{\boldsymbol{m},\boldsymbol{z}}[L\|\epsilon \boldsymbol{v}\|] = L\epsilon \mathbb{E}[\|\boldsymbol{v}\|]. \end{aligned}$$

Since $\boldsymbol{v} = \boldsymbol{m} \odot \boldsymbol{z}$, we have $\mathbb{E}[\|\boldsymbol{v}\|] \leq \sqrt{\mathbb{E}[\|\boldsymbol{v}\|^2]} \approx \sqrt{d}$. Thus, $\text{Bias}^2 \leq L^2 \epsilon^2 d = \mathcal{O}(d\epsilon^2)$. Substituting this back into the above expression yields

$$\|\nabla \mathcal{L}(\boldsymbol{\omega})\|^2 \leq 2\|\nabla \mathcal{L}_\pi(\boldsymbol{\omega})\|^2 + \mathcal{O}(d\epsilon^2). \qquad (72)$$

# G. Proof of Theorem 3.8

**Properties of Smoothed Loss $\mathcal{L}_\pi$.** Recall that in ZO optimization, we optimize the following smoothed objective:

$$\mathcal{L}_\pi(\boldsymbol{\omega}) = \mathbb{E}_{\boldsymbol{m},\boldsymbol{z}}[\mathcal{L}(\boldsymbol{\omega} + \epsilon \boldsymbol{v})], \qquad (73)$$

where $\boldsymbol{v} = \boldsymbol{m} \odot \boldsymbol{z}$ denotes the sparse perturbation.

Under Assumption 3.5, the smoothed loss function $\mathcal{L}_\pi(\boldsymbol{\omega})$ is also $L$-smooth.

Since differentiation and expectation can be interchanged under standard regularity conditions, the gradient of the smoothed loss satisfies $\nabla \mathcal{L}_\pi(\boldsymbol{\omega}) = \mathbb{E}_{\boldsymbol{m},\boldsymbol{z}}[\nabla \mathcal{L}(\boldsymbol{\omega} + \epsilon \boldsymbol{v})]$.

For any $\boldsymbol{\omega}_1, \boldsymbol{\omega}_2$, we have:

$$\begin{aligned} \|\nabla &\mathcal{L}_\pi(\boldsymbol{\omega}_1) - \nabla \mathcal{L}_\pi(\boldsymbol{\omega}_2)\| \\ &= \|\mathbb{E}_{\boldsymbol{m},\boldsymbol{z}}[\nabla \mathcal{L}(\boldsymbol{\omega}_1 + \epsilon \boldsymbol{v}) - \nabla \mathcal{L}(\boldsymbol{\omega}_2 + \epsilon \boldsymbol{v})]\| \\ &\leq \mathbb{E}_{\boldsymbol{m},\boldsymbol{z}}[\|\nabla \mathcal{L}(\boldsymbol{\omega}_1 + \epsilon \boldsymbol{v}) - \nabla \mathcal{L}(\boldsymbol{\omega}_2 + \epsilon \boldsymbol{v})\|] \qquad (74) \\ &\leq \mathbb{E}_{\boldsymbol{m},\boldsymbol{z}}[L\|(\boldsymbol{\omega}_1 + \epsilon \boldsymbol{v}) - (\boldsymbol{\omega}_2 + \epsilon \boldsymbol{v})\|] \\ &= L\|\boldsymbol{\omega}_1 - \boldsymbol{\omega}_2\|. \end{aligned}$$

Thus, $\mathcal{L}_\pi$ is $L$-smooth.

**Lemma G.1** (Approximate Unbiasedness). *Under Assumption 3.6, the sparse estimator $\tilde{\boldsymbol{g}}$ is unbiased for the smoothed gradient $\nabla \mathcal{L}_\pi(\boldsymbol{\omega})$ up to $\mathcal{O}(\epsilon^2)$ terms:*

$$\mathbb{E}_{\boldsymbol{m},\boldsymbol{z}}[\tilde{\boldsymbol{g}}] = \nabla \mathcal{L}_\pi(\boldsymbol{\omega}) + \mathcal{O}(\epsilon^2). \qquad (75)$$

*Proof.* Recall that Proposition 3.2 establishes that the estimator recovers the true gradient up to a second-order bias:

$$\mathbb{E}_{\boldsymbol{m},\boldsymbol{z}}[\tilde{\boldsymbol{g}}] = \nabla \mathcal{L}(\boldsymbol{\omega}) + \mathcal{O}(\epsilon^2). \qquad (76)$$

We next evaluate the gradient of the smoothed loss $\nabla \mathcal{L}_\pi(\boldsymbol{\omega}) = \mathbb{E}_{\boldsymbol{m},\boldsymbol{z}}[\nabla \mathcal{L}(\boldsymbol{\omega} + \epsilon \boldsymbol{v})]$. Under Assumption 3.6, we apply a Taylor expansion to $\nabla \mathcal{L}(\boldsymbol{\omega} + \epsilon \boldsymbol{v})$ at $\boldsymbol{\omega}$:

$$\nabla \mathcal{L}(\boldsymbol{\omega} + \epsilon \boldsymbol{v}) = \nabla \mathcal{L}(\boldsymbol{\omega}) + \epsilon \nabla^2 \mathcal{L}(\boldsymbol{\omega}) \boldsymbol{v}^\top + \mathcal{O}(\epsilon^2 \|\boldsymbol{v}\|^2). \qquad (77)$$

Taking the expectation over $\boldsymbol{v} = \boldsymbol{m} \odot \boldsymbol{z}$, we have

$$\begin{aligned} \nabla \mathcal{L}_\pi(\boldsymbol{\omega}) &= \mathbb{E}_{\boldsymbol{m},\boldsymbol{z}}[\nabla \mathcal{L}(\boldsymbol{\omega})] + \mathbb{E}_{\boldsymbol{m},\boldsymbol{z}}[\epsilon \nabla^2 \mathcal{L}(\boldsymbol{\omega}) \boldsymbol{v}^\top] \\ &\quad + \mathbb{E}_{\boldsymbol{m},\boldsymbol{z}}[\mathcal{O}(\epsilon^2 \|\boldsymbol{v}\|^2)] \\ &= \nabla \mathcal{L}(\boldsymbol{\omega}) + \epsilon \nabla^2 \mathcal{L}(\boldsymbol{\omega}) \mathbb{E}[\boldsymbol{v}^\top] + \mathcal{O}(\epsilon^2) \\ &= \nabla \mathcal{L}(\boldsymbol{\omega}) + \mathcal{O}(\epsilon^2). \end{aligned} \qquad (78)$$

Note that $\mathbb{E}[\boldsymbol{v}^\top] = \boldsymbol{0}^\top$ follows from the independence of $\boldsymbol{m}$ and the symmetry of $\boldsymbol{z} \sim \mathcal{N}(0, \boldsymbol{I}_d)$.

Combining Eq. (76) and Eq. (78), we subtract the two expressions to obtain

$$\begin{aligned} \mathbb{E}[\tilde{\boldsymbol{g}}] - \nabla \mathcal{L}_\pi(\boldsymbol{\omega}) &= (\mathbb{E}[\tilde{\boldsymbol{g}}] - \nabla \mathcal{L}(\boldsymbol{\omega})) - (\nabla \mathcal{L}_\pi(\boldsymbol{\omega}) - \nabla \mathcal{L}(\boldsymbol{\omega})) \\ &= \mathcal{O}(\epsilon^2) - \mathcal{O}(\epsilon^2) \\ &= \mathcal{O}(\epsilon^2). \end{aligned}$$
$$(79)$$

Therefore, $\mathbb{E}_{\boldsymbol{m},\boldsymbol{z}}[\tilde{\boldsymbol{g}}] = \nabla \mathcal{L}_\pi(\boldsymbol{\omega}) + \mathcal{O}(\epsilon^2)$. □

**Proof of Theorem 3.8.** Let $\boldsymbol{\omega}_t$ denote the parameters produced by CurvZO with a fixed learning rate $\eta = \frac{1}{3L}$. Let $F_{ii}(\boldsymbol{\omega}_t)$ denote the $i$-th diagonal entry of the Fisher information matrix evaluated at $\boldsymbol{\omega}_t$.

The proof proceeds by analyzing descent on the smoothed objective and then bounding the discrepancy between the smoothed and original landscapes. We denote $\tilde{\boldsymbol{g}}^t \triangleq \tilde{\boldsymbol{g}}(\boldsymbol{\omega}_t)$ as the sparse ZO gradient computed at iteration $t$.

**Step 1: One-step Descent on Smoothed Loss.**
By the $L$-smoothness of $\mathcal{L}_\pi$, we have:

$$\mathcal{L}_\pi(\boldsymbol{\omega}_{t+1}) \le \mathcal{L}_\pi(\boldsymbol{\omega}_t) + \langle \nabla\mathcal{L}_\pi(\boldsymbol{\omega}_t), \boldsymbol{\omega}_{t+1} - \boldsymbol{\omega}_t \rangle \\ + \frac{L}{2}\|\boldsymbol{\omega}_{t+1} - \boldsymbol{\omega}_t\|^2. \tag{80}$$

Substituting the update rule $\boldsymbol{\omega}_{t+1} = \boldsymbol{\omega}_t - \eta\tilde{\boldsymbol{g}}^t$ yields

$$\mathcal{L}_\pi(\boldsymbol{\omega}_{t+1}) \le \mathcal{L}_\pi(\boldsymbol{\omega}_t) - \eta\langle \nabla\mathcal{L}_\pi(\boldsymbol{\omega}_t), \tilde{\boldsymbol{g}}^t \rangle + \frac{L\eta^2}{2}\|\tilde{\boldsymbol{g}}^t\|^2. \tag{81}$$

We take the conditional expectation with respect to $\boldsymbol{\omega}_t$, denoted as $\mathbb{E}_{\boldsymbol{\omega}_t}[\cdot]$. Using the unbiasedness property $\mathbb{E}_{\boldsymbol{\omega}_t}[\tilde{\boldsymbol{g}}^t] = \nabla\mathcal{L}_\pi(\boldsymbol{\omega}_t) + \mathcal{O}(\epsilon^2)$:

$$\mathbb{E}_{\boldsymbol{\omega}_t}[\mathcal{L}_\pi(\boldsymbol{\omega}_{t+1})] \le \mathcal{L}_\pi(\boldsymbol{\omega}_t) - \eta\langle \nabla\mathcal{L}_\pi(\boldsymbol{\omega}_t), \mathbb{E}_{\boldsymbol{\omega}_t}[\tilde{\boldsymbol{g}}^t] \rangle \\ + \frac{L\eta^2}{2}\mathbb{E}_{\boldsymbol{\omega}_t}[\|\tilde{\boldsymbol{g}}^t\|^2] \\ = \mathcal{L}_\pi(\boldsymbol{\omega}_t) - \eta\langle \nabla\mathcal{L}_\pi(\boldsymbol{\omega}_t), \nabla\mathcal{L}_\pi(\boldsymbol{\omega}_t) + \mathcal{O}(\epsilon^2) \rangle \\ + \frac{L\eta^2}{2}\mathbb{E}_{\boldsymbol{\omega}_t}[\|\tilde{\boldsymbol{g}}^t\|^2]. \tag{82}$$

To handle the cross term $\langle \nabla\mathcal{L}_\pi(\boldsymbol{\omega}_t), \mathcal{O}(\epsilon^2) \rangle$, we use the inequality $ab \le \frac{a^2}{2} + \frac{b^2}{2}$. Let $a = \sqrt{\eta}\|\nabla\mathcal{L}_\pi(\boldsymbol{\omega}_t)\|$ and $b = \sqrt{\eta}\mathcal{O}(\epsilon^2)$. Then,

$$\eta\langle \nabla\mathcal{L}_\pi(\boldsymbol{\omega}_t), \mathcal{O}(\epsilon^2) \rangle \le \frac{\eta}{2}\|\nabla\mathcal{L}_\pi(\boldsymbol{\omega}_t)\|^2 + \frac{\eta}{2}\mathcal{O}(\epsilon^4). \tag{83}$$

Substituting this back yields

$$\mathbb{E}_{\boldsymbol{\omega}_t}[\mathcal{L}_\pi(\boldsymbol{\omega}_{t+1})] \le \mathcal{L}_\pi(\boldsymbol{\omega}_t) - \eta\|\nabla\mathcal{L}_\pi(\boldsymbol{\omega}_t)\|^2 + \frac{\eta}{2}\|\nabla\mathcal{L}_\pi(\boldsymbol{\omega}_t)\|^2 \\ + \frac{\eta}{2}\mathcal{O}(\epsilon^4) + \frac{L\eta^2}{2}\mathbb{E}_{\boldsymbol{\omega}_t}[\|\tilde{\boldsymbol{g}}^t\|^2] \\ = \mathcal{L}_\pi(\boldsymbol{\omega}_t) - \frac{\eta}{2}\|\nabla\mathcal{L}_\pi(\boldsymbol{\omega}_t)\|^2 + \eta\mathcal{O}(\epsilon^4) \\ + \frac{L\eta^2}{2}\mathbb{E}_{\boldsymbol{\omega}_t}[\|\tilde{\boldsymbol{g}}^t\|^2]. \tag{84}$$

**Step 2: Bounding the Second Moment** $\mathbb{E}[\|\tilde{\boldsymbol{g}}^t\|^2]$.
From the variance bound in Eq. (14), we have $\mathbb{E}[(\tilde{\boldsymbol{g}}_i^t)^2] = \mathcal{O}(\frac{F_{ii}}{\pi_i})$. Summing over coordinates yields

$$\mathbb{E}_{\boldsymbol{\omega}_t}[\|\tilde{\boldsymbol{g}}^t\|^2] = \sum_{i=1}^d \mathbb{E}[(\tilde{\boldsymbol{g}}_i^t)^2] = \sum_{i=1}^d \mathcal{O}\left(\frac{F_{ii}}{\pi_i^t}\right). \tag{85}$$

Under the budget constraint $\sum_i \pi_i^t = B$, Proposition 3.4 shows that the variance is minimized when $\pi_i \propto \sqrt{F_{ii}}$.

Substituting this optimal sampling distribution, we obtain

$$\sum_{i=1}^d \frac{F_{ii}}{\pi_i} = \sum_{j=1}^d \left( \frac{F_{jj}}{B\frac{\sqrt{F_{jj}}}{\sum_k \sqrt{F_{kk}}}} \right) \\ = \frac{1}{B}\left( \sum_{k=1}^d \sqrt{F_{kk}} \right)\left( \sum_{j=1}^d \sqrt{F_{jj}} \right) \\ = \frac{1}{B}\left( \sum_{i=1}^d \sqrt{F_{ii}} \right)^2. \tag{86}$$

Therefore, $\mathbb{E}[\|\tilde{\boldsymbol{g}}^t\|^2] \le \frac{C}{B}\left( \sum_{i=1}^d \sqrt{F_{ii}} \right)^2$. Using the assumption $\sum\sqrt{F_{ii}} \le M$, we have

$$\mathbb{E}_{\boldsymbol{\omega}_t}[\|\tilde{\boldsymbol{g}}^t\|^2] \le \frac{CM^2}{B}. \tag{87}$$

Substituting this bound into Eq. (84) yields

$$\mathbb{E}_{\boldsymbol{\omega}_t}[\mathcal{L}_\pi(\boldsymbol{\omega}_{t+1})] \le \mathcal{L}_\pi(\boldsymbol{\omega}_t) - \frac{\eta}{2}\|\nabla\mathcal{L}_\pi(\boldsymbol{\omega}_t)\|^2 \\ + \eta\mathcal{O}(\epsilon^4) + \frac{L\eta^2 CM^2}{2B}. \tag{88}$$

Taking the total expectation $\mathbb{E}[\cdot]$ on both sides, we have

$$\frac{\eta}{2}\mathbb{E}[\|\nabla\mathcal{L}_\pi(\boldsymbol{\omega}_t)\|^2] \le \mathbb{E}[\mathcal{L}_\pi(\boldsymbol{\omega}_t)] - \mathbb{E}[\mathcal{L}_\pi(\boldsymbol{\omega}_{t+1})] \\ + \eta\mathcal{O}(\epsilon^4) + \frac{L\eta^2 CM^2}{2B}. \tag{89}$$

**Step 3: Telescoping Sum.**
Summing the above inequality from $t = 0$ to $T$ yields

$$\frac{\eta}{2}\sum_{t=0}^T \mathbb{E}[\|\nabla\mathcal{L}_\pi(\boldsymbol{\omega}_t)\|^2] \le \sum_{t=0}^T \left( \mathbb{E}[\mathcal{L}_\pi(\boldsymbol{\omega}_t)] - \mathbb{E}[\mathcal{L}_\pi(\boldsymbol{\omega}_{t+1})] \right) \\ + (T+1)\left[ \eta\mathcal{O}(\epsilon^4) + \frac{L\eta^2 CM^2}{2B} \right]. \tag{90}$$

The sum on the right-hand side telescopes to $\mathcal{L}_\pi(\boldsymbol{\omega}_0) - \mathbb{E}[\mathcal{L}_\pi(\boldsymbol{\omega}_{T+1})]$. Assuming $\mathcal{L}_\pi$ is lower bounded by $\mathcal{L}_\pi^*$, we obtain

$$\frac{\eta}{2}\sum_{t=0}^T \mathbb{E}[\|\nabla\mathcal{L}_\pi(\boldsymbol{\omega}_t)\|^2] \le \mathcal{L}_\pi(\boldsymbol{\omega}_0) - \mathcal{L}_\pi^* \\ + (T+1)\eta\mathcal{O}(\epsilon^4) + \frac{L\eta^2 CM^2(T+1)}{2B}. \tag{91}$$

Then, dividing by $\frac{T+1}{2}\eta$ yields

$$\frac{1}{T+1}\sum_{t=0}^T \mathbb{E}[\|\nabla\mathcal{L}_\pi(\boldsymbol{\omega}_t)\|^2] \le \frac{2(\mathcal{L}_\pi(\boldsymbol{\omega}_0) - \mathcal{L}_\pi^*)}{(T+1)\eta} \\ + 2\mathcal{O}(\epsilon^4) + \frac{L\eta CM^2}{B}. \tag{92}$$

**Step 4: From Smoothed Gradient to True Gradient.**

Applying Lemma 3.7 to the time-averaged bound yields

$$
\min_{0 \leq t \leq T} \mathbb{E}[\|\nabla \mathcal{L}(\boldsymbol{\omega}_t)\|^2] \leq \frac{1}{T+1} \sum_{t=0}^{T} \mathbb{E}[\|\nabla \mathcal{L}(\boldsymbol{\omega}_t)\|^2]
$$
$$
\leq 2 \left( \frac{1}{T+1} \sum_{t=0}^{T} \mathbb{E}[\|\nabla \mathcal{L}_\pi(\boldsymbol{\omega}_t)\|^2] \right) + \mathcal{O}(d\epsilon^2). \tag{93}
$$

Combining with Eq. (92), we have

$$
\min_{0 \leq t \leq T} \mathbb{E}[\|\nabla \mathcal{L}(\boldsymbol{\omega}_t)\|^2] \leq \frac{4(\mathcal{L}_0 - \mathcal{L}^*)}{(T+1)\eta} + \frac{2L\eta CM^2}{B} \tag{94}
$$
$$
+ 4\mathcal{O}(\epsilon^4) + \mathcal{O}(d\epsilon^2).
$$

Finally, substituting the learning rate $\eta = \frac{1}{3L}$, we obtain

$$
\min_{0 \leq t \leq T} \mathbb{E}[\|\nabla \mathcal{L}(\boldsymbol{\omega}_t)\|^2] \leq \frac{12L(\mathcal{L}_0 - \mathcal{L}^*)}{T+1} + \frac{2CM^2}{3B} \tag{95}
$$
$$
+ \mathcal{O}(d\epsilon^2).
$$

## H. Proof Sketches for Propositions 3.9 and 3.10

For unbiasedness, by the definition of block-wise gradient estimator $\tilde{\boldsymbol{g}}_{\mathcal{G}_i}$, we have

$$
\mathbb{E}[\tilde{\boldsymbol{g}}_{\mathcal{G}_i}] = \frac{1}{\pi_i^{\text{blk}}} \mathbb{E}\big[ \Delta \, m_i^{\text{blk}} \boldsymbol{z}_{\mathcal{G}_i} \big]. \tag{96}
$$

By independence across blocks, for the $i$-th block, we have

$$
\mathbb{E}\big[ \Delta \, m_i^{\text{blk}} \boldsymbol{z}_{\mathcal{G}_i} \big] = \mathbb{E}\big[ \langle \boldsymbol{g}_{\mathcal{G}_i}, m_i^{\text{blk}} \boldsymbol{z}_{\mathcal{G}_i} \rangle m_i^{\text{blk}} \boldsymbol{z}_{\mathcal{G}_i} \big]
$$
$$
= \mathbb{E}[m_i^{\text{blk}}] \, \mathbb{E}\big[ \langle \boldsymbol{g}_{\mathcal{G}_i}, \boldsymbol{z}_{\mathcal{G}_i} \rangle \boldsymbol{z}_{\mathcal{G}_i} \big] = \pi_i^{\text{blk}} \, \boldsymbol{g}_{\mathcal{G}_i}, \tag{97}
$$

where we used $(m_i^{\text{blk}})^2 = m_i^{\text{blk}}$ and $\mathbb{E}[\boldsymbol{z}_{\mathcal{G}_i} \boldsymbol{z}_{\mathcal{G}_i}^\top] = \boldsymbol{I}$. Thus $\mathbb{E}[\tilde{\boldsymbol{g}}_{\mathcal{G}_i}] = \boldsymbol{g}_{\mathcal{G}_i}$.

For the variance, we consider the second moment $\mathbb{E}\big[ \|\tilde{\boldsymbol{g}}_{\mathcal{G}_i}\|_2^2 \big]$. By definition,

$$
\|\tilde{\boldsymbol{g}}_{\mathcal{G}_i}\|_2^2 = \frac{\Delta^2}{(\pi_i^{\text{blk}})^2} (m_i^{\text{blk}})^2 \|\boldsymbol{z}_{\mathcal{G}_i}\|_2^2
$$
$$
= \frac{\Delta^2}{(\pi_i^{\text{blk}})^2} m_i^{\text{blk}} \|\boldsymbol{z}_{\mathcal{G}_i}\|_2^2. \tag{98}
$$

Taking expectation and using independence across blocks, the mixed moments of $\Delta$ and $m_i^{\text{blk}} \boldsymbol{z}_{\mathcal{G}_i}$ decompose exactly as in the coordinate-wise analysis, with the scalar Fisher diagonal $F_{jj}$ replaced by the block trace $F_i^{\text{blk}}$. This yields

$$
\mathbb{E}\big[ \|\tilde{\boldsymbol{g}}_{\mathcal{G}_i}\|_2^2 \big] \leq C \frac{F_i^{\text{blk}}}{\pi_i^{\text{blk}}}, \tag{99}
$$

*Table 4.* Hyperparameters used in our experiments. CurvZO and CurvZO+LoRA use the same settings as MeZO and MeZO+LoRA, respectively.

| Experiment | Hyperparameters | Values |
|---|---|---|
| FT | Batch size | 8 |
| | Learning rate | {1e−5, 5e−5} |
| | LR schedule | Constant |
| MeZO | Batch size | {64, 16} |
| | Learning rate $\eta$ | {2e−7, 5e−7, 1e−6} |
| | $\epsilon$ | 1e−3 |
| | LR schedule | Constant |
| MeZO+LoRA | Batch size | {64, 16} |
| | Learning rate $\eta$ | {1e−5, 5e−5} |
| | $\epsilon$ | 1e−3 |
| | LR schedule | Constant |

where $C$ is constant independent of $i$, and hence $\text{Var}(\tilde{\boldsymbol{g}}_{\mathcal{G}_i}) \leq C F_i^{\text{blk}} / \pi_i^{\text{blk}}$. Applying the same Lagrange multiplier argument as in the per-parameter case to minimize $\sum_i F_i^{\text{blk}} / \pi_i^{\text{blk}}$ under $\sum_i \pi_i^{\text{blk}} = B$ directly gives $\pi_i^{\text{blk}\star} \propto \sqrt{F_i^{\text{blk}}}$.

## I. Experimental Settings

### I.1. Datasets and Evaluation

We evaluate on SuperGLUE (Wang et al., 2019), including RTE (Dagan et al., 2005), CB (De Marneffe et al., 2019), BoolQ (Clark et al., 2019), WIC (Pilehvar & Camacho-Collados, 2019), and WSC (Levesque et al., 2012). We additionally consider SST-2 (Socher et al., 2013) and two question answering datasets, SQuAD (Rajpurkar et al., 2016) and DROP (Dua et al., 2019).

Following prior work (Malladi et al., 2023; Gao et al., 2021), we randomly sample 1,000 instances for training, 500 for validation, and 1,000 for testing. All experiments are conducted on NVIDIA L40 and A100 GPUs, and results are averaged over three independent runs.

### I.2. Baselines

We compare primarily against two representative ZO optimization methods: memory-efficient ZO optimization (MeZO) (Malladi et al., 2023) and Divergence-driven ZO optimization (DiZO) (Tan et al., 2026). MeZO is a foundational approach for ZO-based LLM fine-tuning, but its full-parameter perturbation strategy results in high estimator variance and slow convergence. DiZO is a divergence-driven zeroth-order optimizer that performs layer-wise adaptation by projecting and scaling ZO updates per layer to better match first-order update behavior, improving convergence and downstream accuracy. In the appendix, we additionally compare with recent ZO baselines, including Sparse-MeZO

*Table 5.* Additional comparison with recent ZO baselines on OPT-2.7B fine-tuning with 1,000 training samples. We report results on six classification tasks and two generation tasks. SensZOQ and MeZO-BCD are only evaluated in the full-parameter ZO setting because their methods do not support LoRA adaptation. The best result among comparable ZO methods in each setting is highlighted in **bold**.

| Method | SST-2 | RTE | CB | BoolQ | WSC | WIC | SQuAD | DROP | Avg. |
|---|---|---|---|---|---|---|---|---|---|
| | | | —classification— | | | | —generation— | | |
| Sparse-MeZO | 92.6 | 66.8 | 69.6 | 62.8 | 61.5 | 55.2 | 79.6 | 24.9 | 64.1 |
| SensZOQ | 90.5 | 60.6 | 64.2 | 64.8 | 54.2 | 55.8 | 72.9 | 25.1 | 61.0 |
| MeZO-BCD | 93.5 | 64.9 | 67.8 | 66.0 | 57.6 | 57.0 | 80.4 | 23.5 | 63.8 |
| SubZero | 85.8 | 54.4 | **71.0** | 65.3 | 60.5 | 52.5 | 61.7 | 18.4 | 58.7 |
| CurvZO | **94.1** | **67.7** | 69.8 | **68.9** | **62.9** | **61.9** | **81.5** | **27.6** | **66.8** |
| Sparse-MeZO+LoRA | 93.1 | 57.4 | 62.5 | 65.2 | 53.8 | 54.2 | 80.8 | 24.4 | 61.4 |
| SubZero+LoRA | 82.6 | 54.2 | 59.1 | 64.8 | 51.4 | 53.2 | 62.4 | 17.2 | 55.6 |
| CurvZO+LoRA | **93.6** | **61.7** | **69.6** | **66.1** | **63.4** | **57.6** | **81.1** | **27.6** | **65.1** |

*Table 6.* Accuracy (%) on OPT-6.7B fine-tuning with different numbers of training samples. Results with 1k examples follow the standard ZO fine-tuning setting, while CurvZO is further evaluated with 2k and 3k examples.

| Method | SST-2 | RTE | WIC | WSC | SQuAD |
|---|---|---|---|---|---|
| | | —classification— | | | -generation- |
| CurvZO (1k) | 94.7 | 72.2 | 61.7 | 61.5 | 83.7 |
| CurvZO (2k) | **94.8** | 73.8 | 63.1 | 63.4 | 84.2 |
| CurvZO (3k) | 94.5 | **79.0** | 64.4 | 63.6 | **84.9** |
| CurvZO+LoRA (1k) | 93.1 | 63.5 | **61.5** | 63.4 | 80.6 |
| CurvZO+LoRA (2k) | **93.6** | 64.2 | 59.0 | **64.2** | **80.9** |
| CurvZO+LoRA (3k) | 93.3 | **64.6** | 59.9 | 62.4 | 80.3 |

*Table 7.* Results on Llama3.1-8B across challenging instruction-following and reasoning benchmarks, including Dolly-15k and GSM8K, as well as standard NLP tasks. Dolly-15k is evaluated by ROUGE-L F1, GSM8K by exact-match accuracy on the final answer, and other tasks by their standard metrics. LoRA is a first-order backpropagation-based baseline, while MeZO and CurvZO are zeroth-order methods. The best ZO results are highlighted in **bold**.

| Method | Dolly-15k | GSM8K | SST-2 | BoolQ | WSC | SQuAD |
|---|---|---|---|---|---|---|
| | –instruction/reasoning– | | —standard NLP tasks— | | | |
| LoRA | 30.7 | 75.7 | 94.5 | 80.1 | 83.5 | 90.7 |
| MeZO | 23.8 | 70.2 | 93.6 | 73.6 | 74.8 | 89.0 |
| CurvZO | **25.4** | **72.4** | **94.4** | **76.1** | **81.8** | **90.3** |

(Liu et al., 2026), SensZOQ (Guo et al., 2025), MeZO-BCD (Park et al., 2025), and SubZero (Yu et al., 2025). These methods improve ZO fine-tuning from different perspectives, such as sparse perturbation, sensitivity-aware quantization, block coordinate descent, and parameter subset selection.

### I.3. Hyperparameter Settings

As shown in Table 4, we largely follow the standard MeZO setup, including the dataset splits, batch size, number of epochs, perturbation scale $\epsilon$, and task prompts. We fix the training length to 20,000 update steps for both OPT and Llama models.

## J. Additional Experiments

**Comparison with additional ZO baselines.** To further evaluate the effectiveness of CurvZO, we include additional comparisons with several recent zeroth-order optimization baselines, including Sparse-MeZO, SensZOQ, MeZO-BCD, and SubZero. For LoRA-based experiments, we compare with the LoRA-compatible baselines, Sparse-MeZO+LoRA and SubZero+LoRA, while SensZOQ and MeZO-BCD are excluded as they do not support LoRA.

As shown in Table 5, CurvZO achieves the best average performance among all compared full-parameter ZO methods. CurvZO also obtains the best results on most individual

tasks, including SST-2, RTE, BoolQ, WSC, WIC, SQuAD, and DROP. This demonstrates that the curvature-guided sparse perturbation strategy remains effective when compared with stronger and more recent ZO baselines.

In the LoRA setting, CurvZO+LoRA consistently outperforms Sparse-MeZO+LoRA and SubZero+LoRA, achieving the highest average score of 65.1. The improvement is especially clear on CB, WSC, WIC, and DROP, suggesting that CurvZO is complementary to parameter-efficient adaptation. Overall, these additional results further support the robustness and generality of CurvZO across both full-parameter and LoRA-based ZO fine-tuning settings.

**Results with larger training sets.** Following prior ZO fine-tuning work, our main experiments adopt the standard 1k-example setting for direct comparison. To further examine whether CurvZO remains effective beyond this commonly used low-data regime, we additionally evaluate OPT-6.7B with larger training sets of 2k and 3k examples. The results are reported in Table 6.

CurvZO generally benefits from larger training sets in full-parameter ZO fine-tuning, with clear improvements on RTE, WIC, WSC, and SQuAD from 1k to 3k examples, while SST-2 remains stable due to near-saturated performance. CurvZO+LoRA also remains competitive under larger data sizes, though with less monotonic gains, suggesting higher

*Table 8.* Fine-tuning results on OPT-2.7B across tasks. **CurvZO** combines curvature-guided sampling with adaptive budget selection. **US+AB** (*Uniform Sampling + Adaptive Budget*) uses the same adaptive budget selection as CurvZO, but samples parameters uniformly at random. **CurvZO** ($B = \rho d$) uses curvature-guided sampling with a fixed perturbation budget $B = \rho d$ (e.g., $\rho \in \{0.1, 0.4, 0.7\}$). The best results in each block are highlighted.

| Task | SST-2 | RTE | CB | BoolQ | WSC | WIC | SQuAD | DROP | Average |
|------|-------|-----|-----|-------|-----|-----|-------|------|---------|
| | | | | —classification— | | | —generation— | | |
| US + AB | 90.8 | 59.9 | 66.7 | 64.2 | 58.6 | 56.5 | 77.8 | 25.1 | 62.5 |
| CurvZO ($B = 0.1d$) | 91.3 | 64.9 | 60.1 | **69.1** | 59.6 | 59.7 | 78.6 | 24.6 | 63.5 |
| CurvZO ($B = 0.4d$) | **94.4** | 67.5 | 67.8 | 65.1 | 56.7 | 60.1 | 80.2 | 24.1 | 64.5 |
| CurvZO ($B = 0.7d$) | 93.0 | 63.8 | 63.2 | 67.3 | 57.6 | 61.1 | 80.0 | 23.1 | 63.6 |
| **CurvZO** | 94.1 | **67.7** | **69.8** | 68.9 | **62.9** | **61.9** | **81.5** | **27.6** | **66.8** |
| US + AB + LoRA | 78.4 | 61.3 | 57.1 | 65.9 | 58.6 | 57.0 | 75.3 | 17.8 | 58.9 |
| CurvZO ($B = 0.1d$) + LoRA | 72.0 | 59.9 | 66.0 | 65.4 | 63.4 | 54.5 | 70.6 | 19.3 | 58.9 |
| CurvZO ($B = 0.4d$) + LoRA | 91.4 | 60.3 | 57.1 | 63.7 | 63.4 | **58.3** | 75.9 | 20.0 | 61.3 |
| CurvZO ($B = 0.7d$) + LoRA | 86.9 | 61.0 | 60.7 | 65.3 | 63.4 | 55.3 | 75.4 | 20.8 | 61.1 |
| **CurvZO + LoRA** | **93.6** | **61.7** | **69.6** | **66.1** | 63.4 | 57.6 | **81.1** | **27.6** | **65.1** |

*Table 9.* Ablation results on OPT-2.7B with 1,000 training samples. We study the effect of EMA smoothing and compare the default $\sqrt{S}$-based sampling rule with an alternative $S$-based rule. The best results in each block are highlighted in **bold**.

| Task | SST-2 | RTE | CB | BoolQ | WSC | WIC | SQuAD | DROP | Average |
|------|-------|-----|-----|-------|-----|-----|-------|------|---------|
| | | | | —classification— | | | —generation— | | |
| CurvZO (w/o EMA) | 90.3 | 59.9 | 67.8 | 67.5 | 51.6 | 60.0 | 77.4 | 25.4 | 62.5 |
| CurvZO ($S$-based sampling) | 92.8 | **70.8** | 69.6 | 61.5 | 61.5 | 58.4 | 81.2 | 25.0 | 65.1 |
| CurvZO | **94.1** | 67.7 | **69.8** | **68.9** | **62.9** | **61.9** | **81.5** | **27.6** | **66.8** |
| CurvZO (w/o EMA)+LoRA | 92.3 | 56.3 | 66.0 | 64.3 | 50.9 | 55.8 | 79.7 | 24.4 | 61.2 |
| CurvZO ($S$-based sampling)+LoRA | 93.5 | 57.4 | 60.7 | 65.1 | 60.3 | 56.8 | 78.9 | 24.4 | 62.1 |
| CurvZO+LoRA | **93.6** | **61.7** | **69.6** | **66.1** | **63.4** | **57.6** | **81.1** | **27.6** | **65.1** |

sensitivity to task difficulty, adapter capacity, and data size. Overall, the results confirm that CurvZO remains effective beyond the standard 1k-example regime.

**Results on Llama3.1-8B.** To further evaluate CurvZO on more recent LLMs and more challenging tasks, we conduct additional experiments on Llama3.1-8B. In addition to standard classification and generation benchmarks, we include Dolly and GSM8K to assess instruction-following generation and mathematical reasoning abilities, respectively. For Dolly-15k, we report ROUGE-L F1, computed as the best match between the prediction and the reference responses and averaged over examples. For GSM8K, we report exact-match accuracy on the final answer.

As shown in Table 7, CurvZO consistently outperforms MeZO across all evaluated tasks. Although first-order LoRA remains stronger on Dolly-15k and GSM8K, CurvZO substantially narrows the gap while preserving the memory advantages of ZO optimization. These results suggest that curvature-guided sparse perturbations remain effective on newer model architectures and extend to more challenging instruction-following and reasoning benchmarks.

# K. Ablation Experiments

Table 8 reports an ablation study on OPT-2.7B across classification and generation tasks. We evaluate the contribution of the two key components in CurvZO: (i) curvature-guided sparse ZO (i.e., using the curvature scores to select parameters for perturbation) and (ii) adaptive budget selection. As a baseline, **US+AB** uses the same adaptive budget selection as CurvZO but samples perturbed parameters uniformly at random, which yields noticeably worse performance than CurvZO, highlighting the importance of curvature-guided sampling. We further test **CurvZO** ($B = \rho d$) variants that use curvature-guided sampling with a fixed perturbation budget. Across most tasks, **CurvZO outperforms the fixed-budget variants**, indicating the benefit of adaptive budget selection. The same trend holds under the LoRA setting: CurvZO+LoRA achieves the best overall average, outperforming both US+AB+LoRA and fixed-budget CurvZO+LoRA variants. Overall, the results confirm that curvature-guided sampling and adaptive budget selection are complementary, and combining them yields the strongest and most robust gains.

We further conduct ablation studies on OPT-2.7B to examine two design choices in CurvZO: EMA smoothing for the online curvature scores and the $\sqrt{S}$-based sampling rule.

The variant without EMA directly uses raw online curvature scores for sampling, while the $S$-based variant samples coordinates proportionally to $S$ instead of the default $\sqrt{S}$ rule. As shown in Table 9, removing EMA consistently degrades the average performance in both full-parameter and LoRA-based ZO fine-tuning. The average score drops from 66.8 to 62.5 in the full-parameter setting and from 65.1 to 61.2 in the LoRA setting, indicating that EMA helps stabilize noisy online curvature estimates and prevents the sampling distribution from being dominated by transient perturbation noise. The comparison between $\sqrt{S}$-based and $S$-based sampling further validates our sampling design. Although $S$-based sampling performs well on some individual tasks, such as RTE in the full-parameter setting, it yields lower average performance than the default CurvZO in both settings. This suggests that directly sampling according to $S$ can over-concentrate the perturbation budget on a small set of high-score coordinates, whereas the $\sqrt{S}$ rule provides a better balance between exploiting high-curvature directions and maintaining sufficient exploration. Overall, these ablations support the use of EMA-smoothed curvature scores and the $\sqrt{S}$-based sampling rule in CurvZO.

## L. Pseudocode

Algorithm 1 summarizes the block-wise implementation of CurvZO.

---

**Algorithm 1** CurvZO: Block-wise Version

---

**Require:** Initial parameters $\boldsymbol{\omega}_0$; block partition $\{\mathcal{G}_i\}_{i=1}^{G}$; perturbation scale $\epsilon$; learning rate $\eta$; EMA coefficient $\beta$; budget range $[B_{\min}, B_{\max}]$; initial budget $B_{\text{init}}$; budget update interval $K_B$; trade-off coefficient $\alpha$.

1: Initialize block-wise curvature scores $S_i^{\text{blk},0} \leftarrow 1$, $\forall i \in [G]$, and $B \leftarrow B_{\text{init}}$.
2: **for** $t = 0, \ldots, T - 1$ **do**
3:     **if** $t \bmod K_B = 0$ **then**
4:         Compute $d_{\text{eff}}$ and $H$ from the block-wise score distribution $\{S_i^{\text{blk},t}\}_{i=1}^{G}$, and update the perturbation budget using Eq. (20):
$$B \leftarrow B_{\min} + (B_{\max} - B_{\min}) \left( \alpha \frac{d_{\text{eff}}}{G} + (1 - \alpha) H \right).$$

5:     **end if**
6:     Set block sampling probabilities:
$$\pi_i^{\text{blk},t} \propto \sqrt{S_i^{\text{blk},t}}, \qquad \sum_{i=1}^{G} \pi_i^{\text{blk},t} = B, \qquad 0 < \pi_i^{\text{blk},t} \leq 1.$$

7:     Sample a minibatch $\mathcal{B}_t$.
8:     Sample block-wise sparse perturbation:
$$m_i^{\text{blk},t} \sim \text{Bernoulli}(\pi_i^{\text{blk},t}), \qquad \boldsymbol{v}_{\mathcal{G}_i}^{\text{blk},t} = m_i^{\text{blk},t} \boldsymbol{z}_{\mathcal{G}_i}^t, \quad \boldsymbol{z}_{\mathcal{G}_i}^t \sim \mathcal{N}(\boldsymbol{0}, \boldsymbol{I}_{|\mathcal{G}_i|}).$$

9:     Compute the two-point ZO response on the same minibatch:
$$\Delta_t = \frac{\mathcal{L}(\boldsymbol{\omega}_t + \epsilon \boldsymbol{v}^{\text{blk},t}; \mathcal{B}_t) - \mathcal{L}(\boldsymbol{\omega}_t - \epsilon \boldsymbol{v}^{\text{blk},t}; \mathcal{B}_t)}{2\epsilon}.$$

10:    Estimate the block-wise ZO gradient:
$$\tilde{\boldsymbol{g}}_{\mathcal{G}_i}^t = \Delta_t \frac{\boldsymbol{v}_{\mathcal{G}_i}^{\text{blk},t}}{\pi_i^{\text{blk},t}}, \qquad i = 1, \ldots, G.$$

11:    Update block-wise curvature scores:
$$\tilde{s}_i^{\text{blk},t} = \frac{\|\boldsymbol{v}_{\mathcal{G}_i}^{\text{blk},t}\|_2^2}{\|\boldsymbol{v}^{\text{blk},t}\|_2^2} \Delta_t^2, \qquad S_i^{\text{blk},t+1} = (1 - \beta) S_i^{\text{blk},t} + \beta \tilde{s}_i^{\text{blk},t}.$$

12:    Update parameters:
$$\boldsymbol{\omega}_{t+1} = \boldsymbol{\omega}_t - \eta \tilde{\boldsymbol{g}}^t.$$

13: **end for**
14: **return** $\boldsymbol{\omega}_T$

---

