# OpenReview forum: "CurvZO: Adaptive Curvature-Guided Sparse Zeroth-Order Optimization for Efficient LLM Fine-Tuning"
_ICML.cc/2026/Conference — ICML 2026 regular_

### Official Review · Reviewer_a2X2 · 2026-02-25

**Soundness:** 3
**Presentation:** 1
**Significance:** 2
**Originality:** 3
**Overall Recommendation:** 4
**Confidence:** 5

**Summary:**

This paper introduces CurvZO, an adaptive curvature-guided sparse zeroth-order optimization algorithm for efficient fine-tuning of large language models. The method leverages curvature information from the loss function to guide sparse parameter updates, avoiding the need to compute full gradients as in traditional zeroth-order optimization methods. CurvZO adaptively selects important parameters for updates, significantly reducing computational overhead while maintaining fine-tuning effectiveness.

**Compliance With Llm Reviewing Policy:**

Affirmed.

**Final Justification:**

The rebuttal has addressed all of the specific questions I raised. However, as noted by other reviewers, the manuscript still exhibits certain flaws, including insufficient experimentation and the configuration of EMA. In my assessment, there remains a significant gap before the paper reaches a score of 5. Therefore, I will stand by my original score (4 weak accept).

**Key Questions For Authors:**

It is recommended that the authors prioritize resolving the concerns outlined under Weakness before proceeding to respond to the questions below.

1. What is the sensitivity of the method to the sparsity ratio and curvature threshold? Are there guidelines for setting these hyperparameters?

2. Can you analyze the approximation error introduced by sparse updates and its impact on optimization dynamics?

**Limitations:**

See weaknesses.

**Strengths And Weaknesses:**

**Strengths**

1. The combination of curvature-guided selection with sparse zeroth-order optimization addresses a practical need for memory-efficient LLM fine-tuning, particularly when gradient computation is expensive or infeasible.

2. The adaptive parameter selection mechanism based on curvature information is a principled approach that could potentially identify more important parameters than simple magnitude-based methods.

3. The work tackles an underexplored area—sparse zeroth-order optimization for LLMs—which could be valuable for scenarios with limited computational resources or when gradients are unavailable.

**Weaknesses**

1. The theoretical analysis appears limited. The convergence guarantees and the relationship between curvature estimation accuracy and optimization performance need more rigorous treatment. How does the approximation error in curvature estimation affect the final solution quality?

2. The experimental evaluation lacks comprehensive comparisons with relevant baselines. How does CurvZO compare against other sparse optimization methods (e.g., SparseMEZO)

3. The absence of pseudocode in this article makes it challenging to rapidly understand the algorithm’s innovative aspects.

4. The authors seem to keep an exponential moving average for every parameter (Eq. 8), but no associated growth in memory cost is observed, creating an inconsistency (and in the absence of pseudocode, the precise algorithmic implementation remains unclear).

5. The main experiments in the paper focus on relatively outdated models such as OPT and LLaMA2, as well as the SuperGLUE dataset; incorporating more recent benchmarks and newer models such as LLaMA3 and Qwen3 would help to more thoroughly assess the method’s contribution.

---

> ### Author Rebuttal · Authors · 2026-03-31
>
> We thank the reviewer for the constructive feedback.
>
> > Weakness 1
>
> We agree that the convergence bound does not explicitly characterize how curvature-estimation accuracy affects the final solution quality. Nevertheless, the paper still provides rigorous analysis of the core mechanism and offers meaningful theoretical contributions.
>
> - **Challenge in the theory.** The main difficulty comes from the **normalization** and **EMA smoothing** used in the practical curvature score. These stabilize online score tracking, but also make the gap between the Fisher-diagonal-style target and the practical score hard to characterize (see our response to **Weakness 1** from reviewer **VPph**). Consequently, the gap between the practical sampling rule $\ \pi_i^{t} \propto \sqrt{S_i}$  and the ideal Fisher-based rule $ \pi_i^{t\star} \propto \sqrt{F_{ii}}$  is difficult to quantify cleanly, so curvature-estimation accuracy cannot yet be propagated explicitly into the final bound.
> - **Method validity.**  Although a full characterization is still difficult, we do provide explicit analysis of the **one-step curvature score**. Eq. (6) shows that $\mathbb{E}[s_i]$ decomposes into a coordinate-wise term aligned with $g_i^2$ (a standard stochastic proxy for the diagonal Fisher), together with a shared term and the standard ZO approximation error. This already suffices to justify using the score to construct the sampling rule, since it only needs to capture the **relative importance** across coordinates.
> - **Theoretical contribution.** Theorem 3.8 could be further strengthened by explicitly characterizing the impact of curvature-estimation accuracy, but it already provides a nontrivial guarantee. To the best of our knowledge, it is the **first convergence guarantee for sparse ZO fine-tuning with a Fisher-diagonal-guided, time-varying sampling rule**, rather than a fixed sparse subset. By comparison, Sparse-MeZO only provides a convergence guarantee for a fixed sparse subset.
>
> > Weakness 2
>
> To address this, we add several ZO baselines: **Sparse-MeZO, SensZOQ, MeZO-BCD, and SubZero**. See our response to reviewer **x5cB, Weakness 2** for full results.
>
> > Weakness 3
>
> Thank you for this comment. We will add concise pseudocode in the revision to make the execution flow explicit.
>
> > Weakness 4
>
> We would like to clarify that, in our implementation, we use the **block-wise curvature scores** in Section 3.5 rather than per-parameter scores, which is consistent with the experimental results in Table 3. We present the parameter-level formulation first because it gives the clearest explanation of the score construction, bias correction, and variance-guided sampling rule. To make CurvZO **practical** for LLM fine-tuning, we use a block-wise implementation that **reduces memory and per-parameter masking overhead**. In block-wise CurvZO, each block corresponds to one trainable parameter tensor (e.g., a projection matrix or bias vector), so we maintain one EMA score per block. For Llama2-13B, this yields **363 blocks**, so the extra memory cost is only **363 float32 values** (about **1.4 KB**).
>
> > Weakness 5
>
> Our current choice of models and benchmarks follows the standard evaluation setup in closely related prior ZO fine-tuning work, which allows direct and fair comparison with existing methods. We also agree that evaluation on newer models and benchmarks, such as LLaMA3, Qwen, and more recent tasks, would further strengthen the paper. However, due to the limited rebuttal period, we were unable to complete such additional large-scale experiments. We strengthen the empirical evaluation in other ways by adding more baselines and new ablation studies. Detailed results are provided in our responses to **Weakness 2** and **Key Question 3** from reviewer **x5cB**, and to **Weakness 3** from reviewer **VPph**.
>
> > Key Question 1
>
> We would like to clarify that CurvZO does **not** rely on a fixed curvature threshold, nor a manually specified fixed sparsity ratio. Instead, curvature scores are mapped to sampling probabilities, and the perturbation budget is adaptively determined from the score distribution in Section 3.3. The fixed-budget settings in Table 4 are included only as ablations to study the effect of different sparsity levels, while the adaptive budget selection is the default CurvZO configuration and performs best overall.
>
> > Key Question 2
>
> We would like to clarify that sparse updates in CurvZO do **not** introduce additional bias into the optimization dynamics. While the naive sparse estimator $\hat{g}=\Delta v$ is biased under Bernoulli masking (Eqs. (9)–(10)), CurvZO explicitly corrects this using the Horvitz–Thompson estimator $\tilde{g}_i = \Delta v_i /\pi_i$. Proposition 3.2 shows that this estimator is unbiased, so sparse masking does not distort the intended descent direction in expectation. Its impact is instead reflected in the variance of the estimator, rather than in an additional sampling bias in the optimization dynamics.

---

> > ### Author Rebuttal · Reviewer_a2X2 · 2026-04-02
> >
> > I thank the authors for their rebuttal and am pleased to maintain my original positive score.
> >
> > I strongly recommend that the authors evaluate the proposed algorithm on more advanced benchmarks and provide comparisons with FO algorithms regarding memory footprint and runtime. Present ZO optimization literature lacks sufficient validation on these difficult benchmarks, which is essential for demonstrating the algorithm's practicality as a fine-tuning alternative in resource-limited industrial scenarios.

---

> > > ### Author Response · Authors · 2026-04-07
> > >
> > > Thank you for the positive feedback and for maintaining your supportive assessment. We also appreciate the reviewer’s valuable suggestion to further strengthen the empirical validation using more advanced models and more challenging benchmarks. In response, we have expanded the experiments to include a **newer model, Llama3.1-8B**, as well as **more challenging benchmarks, Dolly and GSM8K** (please see our response to **Reviewer 8aw8** for details). We hope these additions further reinforce the practical relevance of CurvZO as a fine-tuning alternative in resource-constrained settings.

---

### Official Review · Reviewer_8aw8 · 2026-03-12

**Soundness:** 3
**Presentation:** 3
**Significance:** 2
**Originality:** 3
**Overall Recommendation:** 3
**Confidence:** 4

**Summary:**

This paper introduces CurvZO, a method designed to improve the convergence and accuracy of memory-efficient zeroth-order (ZO) fine-tuning for LLMs. While existing sparse ZO methods perturb a subset of parameters to reduce gradient variance, they lack a principled way to select informative parameters using only scalar feedback. CurvZO addresses this by tracking local curvature signals online directly from scalar ZO responses to create a sampling distribution, prioritizing perturbations on high-curvature parameters. Furthermore, CurvZO employs an adaptive budget mechanism that dynamically adjusts the number of perturbed parameters based on the evolving sharpness of the curvature distribution. To minimize computational and memory overhead, the framework is implemented at the block level, treating parameter tensors as individual units. Empirical evaluations on OPT and Llama models across various NLP benchmarks show that CurvZO outperforms baselines like MeZO and DiZO, achieving higher accuracy and faster convergence while maintaining standard ZO memory efficiency.

**Compliance With Llm Reviewing Policy:**

Affirmed.

**Final Justification:**

Thank you for the rebuttal and additional experiments. The paper addresses an interesting problem.

However, several concerns remain after the rebuttal. First, the performance gap between CurvZO and LoRA on challenging benchmarks (Dolly, GSM8K) is still substantial, and the claim of being "reasonably competitive" is difficult to accept given the numbers. Second, the wall-clock training time of CurvZO is approximately 15–20× longer than LoRA, which significantly limits practical applicability outside of memory-constrained scenarios. The rebuttal did not sufficiently address the question of when CurvZO would be preferred over LoRA in practice.

The rebuttal partially addressed our original concerns by extending the evaluation to newer models and more challenging benchmarks, which is appreciated. However, it did not change our overall assessment. We maintain our current score.

**Key Questions For Authors:**

Q1.  In the context of Theorem 3.8, how is the theoretical variance explicitly bounded from exploding to infinity, given that no strictly positive lower bound ($\pi_{min} > 0$) is defined for the sampling probabilities? Does the block-wise implementation serve as an implicit lower bound, and if so, can this be formalized in the theory?

Q2. Could the authors explicitly discuss the conceptual and methodological differences between CurvZO and this concurrent low-rank curvature approach? Low-Rank Curvature for Zeroth-Order Optimization in LLM Fine-Tuning (Seung et al., 2025)

**Limitations:**

Yes

**Strengths And Weaknesses:**

**Strength:**

1.The paper addresses a highly relevant and critical problem: the prohibitive memory wall in fine-tuning Large Language Models (LLMs). Transitioning from random sparse zeroth-order (ZO) perturbations to a signal-guided, informative sparsity pattern is a meaningful advancement that provides clear practical utility for resource-constrained LLM adaptation.

2.The work introduces a creative and novel engineering integration. While utilizing Fisher-like information for parameter selection is known in first-order optimization, the authors successfully map this to a purely ZO setting by tracking scalar feedback online. Combining this with an adaptive budget selection based on the evolving sharpness of the signal distribution is a highly original system-level design.

3.The block-wise implementation is a highly pragmatic engineering compromise that successfully bridges the gap between theoretical per-parameter tracking and the massive dimensionality of actual LLMs. The empirical results demonstrate consistent improvements over strong baselines like MeZO and DiZO across multiple tasks and credible model architectures (OPT and Llama up to 13B).

**Weakness:**

1. Experiment Benchmark Methods not enough. While the paper explicitly cites SensZOQ as a closely related sparse ZO method in the motivation, it does not include it as an empirical baseline, leaving the comparative evaluation somewhat incomplete.

2. I have a question, only 1000 training datasets is enough?  The empirical study is confined to a very low-data regime, using only 1,000 training samples per task. For models at the 7B-13B, this constrained setting limits the external validity of the robustness claims and optimization dynamics.

3. For the training speed. Is it possible for author to compared our method to Full FT and Lora.

---

> ### Author Rebuttal · Authors · 2026-03-31
>
> We thank the reviewer for the constructive comments. Our responses are as follows.
>
> > Weakness 1
>
> To address this, we have added several additional ZO baselines, including **SensZOQ, Sparse-MeZO, MeZO-BCD, and SubZero**. We refer the reviewer to our response to reviewer **x5cB, Weakness 2** for the full results and discussion.
>
> > Weakness 2
>
> Our use of **1k training examples per task** follows the standard setting in prior ZO fine-tuning work and is primarily intended to support fair comparison with existing methods. To address the reviewer's question, we additionally include experiments on **OPT-6.7B** with larger training sets (**2k** and **3k** examples). Detailed results and discussion are provided in our response to reviewer **x5cB**, **Key Question 3**.
>
> > Weakness 3
>
> We would like to clarify that FT/LoRA are faster than ZO methods in raw training speed, because ZO must estimate gradients through extra forward evaluations. Our point is not that CurvZO is faster than FT/LoRA, but that it preserves the **low-memory advantage** of ZO while making ZO itself more efficient. As shown in Table 3, CurvZO uses only **5.91 GB** on OPT-2.7B and **13.95 GB** on OPT-6.7B, whereas FT requires **42.88–45.28 GB** and **>80 GB**, respectively. Within this low-memory regime, CurvZO reaches MeZO’s accuracy in **2.4× fewer steps** on BoolQ and **2.0× fewer steps** on RTE, reducing GPU hours by up to **59%** and **51%**. This makes CurvZO especially useful in **single-GPU or limited-memory settings**.
>
> > Key Question 1
>
> **Sampling-probability feasibility condition.** We first clarify that $\pi_i \in (0,1]$ is the default feasibility condition introduced at the beginning of Section 3.
>
> **Variance boundedness without an explicit $\pi_{\text{min}}>0$.** The boundedness in Theorem 3.8 does **not** rely on an explicit lower bound $\pi_{\min}>0$. Instead, it comes from the **variance-minimizing sampling rule** under the budget constraint $\sum_{i=1}^d \pi_i^t = B$. Accordingly, the proof controls the aggregate variance surrogate
> $$
> \sum_{i=1}^d \frac{F_{ii}(\omega_t)}{\pi_i^t}
> $$
> under the optimizer of Eq. (15), rather than under arbitrary sampling probabilities. In the interior case of Proposition 3.4, the minimized value is
> $$
> \sum_{i=1}^d \frac{F_{ii}(\omega_t)}{\pi_i^{t,\star}} = \frac{1}{B}\Big(\sum_{i=1}^d \sqrt{F_{ii}(\omega_t)}\Big)^2 \le \frac{M^2}{B},
> $$
> which is exactly the source of the $\mathcal{O}(M^2/B)$ variance floor in Theorem 3.8.
>
> **Block-wise implementation.** The block-wise implementation does not serve as an implicit lower bound on the sampling probabilities in the sense of introducing a formal $\pi_{\min}>0$ assumption. Instead, it is a scalable aggregation of the same variance-minimizing principle at the block level, with
> $$
> F^{\mathrm{blk}}\_i = \sum_{j \in \mathcal{G}\_i} F_{jj}, \qquad \pi_i^{\mathrm{blk},\star} \propto \sqrt{F_i^{\mathrm{blk}}}.
> $$
> Under $\sum_{i=1}^G \pi_i^{\mathrm{blk}} = B$, the same argument gives
> $$
> \sum_{i=1}^G \frac{F_i^{\mathrm{blk}}(\omega_t)}{\pi_{i,t}^{\mathrm{blk}}} = \frac{1}{B}\Big(\sum_{i=1}^G \sqrt{F_i^{\mathrm{blk}}(\omega_t)}\Big)^2.
> $$
> Moreover,
> $$
> \sum_{i=1}^G \sqrt{F_i^{\mathrm{blk}}(\omega_t)} =\sum_{i=1}^G \sqrt{\sum_{j\in \mathcal{G}\_i} F_{jj}(\omega_t)} \le \sum\_{j=1}^d \sqrt{F_{jj}(\omega_t)} \le M,
> $$
> so the same $\mathcal{O}(M^2/B)$ variance floor carries over to the block-wise setting. Therefore, the block-wise implementation does not justify boundedness by imposing an implicit lower bound on probabilities; instead, it inherits the same aggregate variance control under the block-level curvature-aware allocation rule.
>
> > Key Question 2
>
> Although both methods exploit curvature information in ZO optimization, they differ fundamentally in both concept and methodology.
>
> - **Conceptual difference.** **LOREN** uses curvature to learn a **dense anisotropic perturbation distribution**, with the goal of improving full-dimensional search directions. **CurvZO**, instead, uses curvature to decide **where the limited perturbation budget should be placed**, by allocating higher sampling probabilities to more informative coordinates. This makes CurvZO more directly aligned with the key decision in sparse ZO: **informative budget allocation**, rather than dense perturbation reshaping.
> - **Methodological difference.** **LOREN** introduces an additional **covariance / search-distribution learning procedure** based on ES-style modeling and low-rank covariance updates. **CurvZO** stays within the **standard two-point ZO pipeline** and uses only the scalar response already produced during training to build the sampling rule. This gives CurvZO a practical advantage in **plug-and-play** integration and avoids introducing an extra distribution-learning module.
>
> In summary, CurvZO is better suited to sparse ZO fine-tuning because it directly targets informative budget allocation while preserving the simplicity and low-memory nature of the standard ZO pipeline.

---

> > ### Author Rebuttal · Reviewer_8aw8 · 2026-04-01
> >
> > Thanks the authors for their response, which addresses part of our concerns. We have the following additional questions:
> >
> > 1.The current generation benchmarks (SQuAD, DROP) are essentially short-answer tasks producing only a few tokens. Has CurvZO been evaluated on tasks requiring longer sequence generation (e.g., summarization or instruction following)?
> >
> > 2.How does CurvZO compare with LoRA in terms of training speed, for example reach to the convergence state, what's the GPU hour is.

---

> > > ### Author Response · Authors · 2026-04-07
> > >
> > > Thank you for the helpful follow-up questions. In response, we have strengthened the experiments by adding a **newer model (Llama3.1-8B)** and **more challenging benchmarks (Dolly and GSM8K)**, with detailed results shown in the table below.
> > >
> > > > Q1.
> > >
> > > To broaden the evaluation scope, we additionally include **Dolly** and **GSM8K**. In particular, **Dolly** evaluates instruction-following generation, while **GSM8K** introduces substantially more challenging reasoning, thereby extending the empirical scope beyond the original short-answer setting.
> > >
> > > Extended evaluation on **Llama3.1-8B** with additional challenging benchmarks (**Dolly** and **GSM8K**):
> > >
> > > | Task   | Dolly | GSM8K | SST-2 |  RTE | BoolQ | SQuAD |
> > > | ------ | ----- | ----- | ----: | ---: | ----: | ----: |
> > > | LoRA   | 30.7  | 75.7  |  94.5 | 80.1 |  83.5 |  90.7 |
> > > | MeZO   | 23.8  | 70.2  |  93.6 | 73.6 |  74.8 |  89.0 |
> > > | CurvZO | 25.4  | 72.4  |  94.4 | 76.1 |  81.8 |  90.3 |
> > >
> > > These results show that CurvZO consistently improves over **MeZO** on the newer model and the added challenging benchmarks, while remaining reasonably competitive with **LoRA**.
> > >
> > > > Q2.
> > >
> > > To further address efficiency, we include **wall-clock training time** and **GPU memory** comparisons with **LoRA**, so that the practical trade-off of CurvZO can be evaluated more directly.
> > >
> > > Efficiency comparison on OPT-2.7B. Top: **wall-clock training time** (minutes). Bottom: peak **GPU memory usage**.
> > >
> > > | Task   |     SST-2 |        RTE |       WIC |        WSC |   SQuAD    |
> > > | ------ | --------: | ---------: | --------: | ---------: | :--------: |
> > > | LoRA   |   5.6 min |  17.06 min | 14.62 min |   6.67 min | 18.43 min  |
> > > | CurvZO | 90.89 min | 112.83 min | 87.29 min | 105.20 min | 151.64 min |
> > >
> > > | Method | Memory         | Mem. Ratio vs FT |
> > > | ------ | -------------- | ---------------- |
> > > | FT     | 42.88-45.28 GB | 1.00×            |
> > > | LoRA   | 15.24-17.46 GB | 0.37×            |
> > > | CurvZO | 5.91 GB        | 0.13×            |
> > >
> > > These results clarify the practical trade-off of **CurvZO**: although its wall-clock training time is longer than **LoRA**, it requires substantially less GPU memory. This makes CurvZO particularly valuable in scenarios where backpropagation-based fine-tuning may be difficult or infeasible because the **additional training states required for backpropagation** (e.g., activations, gradients, and optimizer states) cannot fit into limited GPU memory.
> > >
> > > We hope these additional experiments make the empirical evaluation more comprehensive in terms of **task difficulty, model modernity, and practical efficiency**.

---

### Official Review · Reviewer_VPph · 2026-03-12

**Soundness:** 3
**Presentation:** 3
**Significance:** 3
**Originality:** 2
**Overall Recommendation:** 4
**Confidence:** 3

**Summary:**

This paper studies ZO fine-tuning for large language models under memory constraints. The main idea is to use online curvature estimates from scalar ZO feedback to decide which parameters or blocks are perturbed. The method also adapts the perturbation budget during training and includes an inverse-probability correction to reduce bias from sparse masking. The paper provides variance and convergence analysis, and experiments on OPT and Llama models across several NLP tasks show improvements over prior ZO baselines.

**Compliance With Llm Reviewing Policy:**

Affirmed.

**Final Justification:**

The rebuttal has addressed most of my concerns. Considering the importance of the problem and the novelty of the method, I am suggesting weak accept. I cannot give a higher recommendation since both theoretic and experimental parts can still be improved.

**Key Questions For Authors:**

See weaknesses part.

**Limitations:**

Yes.

**Strengths And Weaknesses:**

Strengths:
* The paper addresses a relevant problem: improving the efficiency of ZO fine-tuning for LLM.
* The proposed method is well motivated. Using curvature information to guide sparse perturbations is intuitive and technically reasonable.
* The paper includes both theory and experiments.
* Empirical results are promising, with generally better accuracy and/or faster convergence than prior ZO baselines while keeping memory use close to MeZO.

Weaknesses:
* The analysis is motivated by Fisher-diagonal-style quantities, but the actual method uses a smoothed curvature proxy; the gap between the two is not fully justified.
* The formal guarantees do not explain some of the choices in algorithm, e.g. adaptive budget selection and block-wise implementation.
* Experiments can be stronger with ablation studies.

---

> ### Author Rebuttal · Authors · 2026-03-31
>
> We thank the reviewer for the thoughtful comments. Our responses are provided below.
>
> > Weakness 1
>
> We would like to clarify that the gap between the Fisher-diagonal-style target and the curvature score has been **explicitly characterized at the one-step level**. Specifically, Eq. (6) shows that $\mathbb{E}[s_i]$ decomposes into **(i)** a coordinate-wise term aligned with $g_i^2$ (a standard stochastic proxy for the diagonal Fisher), **(ii)** a shared term, and **(iii)** the standard ZO approximation error. This is already sufficient for constructing the sampling rule, since the score only needs to provide a reliable signal of the **relative importance** across coordinates.
>
> In the practical algorithm, we further introduce **normalization** and **EMA smoothing**:
> $$
> S_i^t=(1-\beta)S_i^{t-1}+\beta\frac{s_i^t}{\sum_jv_j^2},
> $$
> to make the online signal stable enough for training. Although the theoretical analysis would be strengthened by characterizing the gap between this **stabilized practical score** and the Fisher-diagonal-style target, this can be very challenging:
>
> - **Normalization** introduces a random denominator coupled with the numerator;
> - **EMA** makes $S_i^t$ history-dependent rather than a one-step quantity;
>
> Although we are unable to fully characterize this gap during the rebuttal period, the theoretical contribution remains nontrivial. To the best of our knowledge, it is the **first convergence guarantee for sparse ZO fine-tuning with a Fisher-diagonal-guided, time-varying sampling rule**, rather than a fixed sparse subset.
>
> > Weakness 2
>
> We would like to clarify that Theorem 3.8 directly extends to the block-wise implementation, whereas adaptive budget selection is motivated by stable online curvature tracking rather than explicitly covered by the fixed-budget theorem.
>
> For the **block-wise implementation**, the same convergence argument extends directly. Appendix H already gives the block-wise consistency result
> $$
> \mathbb{E}[\tilde{g}\_{\mathcal{G}\_i}]=g_{\mathcal{G}\_i}+\mathcal{O}(\epsilon^2),
> $$
> together with the block-level second-moment bound
> $$
> \mathbb{E} \\| \tilde{g}\_t\^{\mathrm{blk}} \\|\_2\^2 \le C \sum\_{i=1}\^G \frac{F_i^{\mathrm{blk}}(\omega_t)}{\pi_{i,t}\^{\mathrm{blk}}},
> $$
> Under the block budget constraint $\sum_{i=1}^G \pi_{i,t}^{\mathrm{blk}}=B$, the same argument as in the coordinate-wise case gives
> $$
> \pi_{i,t}^{\mathrm{blk},\star}\propto \sqrt{F_i^{\mathrm{blk}}(\omega_t)}, \qquad \mathbb{E}\\|\tilde g_t^{\mathrm{blk}}\\|\_2^2\leq\frac{CM^2}{B},
> $$
> since $\sum_{i=1}^G\sqrt{F_i^{\mathrm{blk}}(\omega_t)}\le\sum_{j=1}^d\sqrt{F_{jj}(\omega_t)}\le M.$ Therefore, the same $\mathcal{O}(M^2/B)$ variance floor holds in the block-wise setting. Theorem 3.8 extends directly to the block-wise implementation.
>
> For **adaptive budget selection**, Theorem 3.8 does not directly cover the adaptive-budget case, since adaptive budget is introduced mainly to stabilize **online curvature-score tracking** rather than to optimize the fixed-$B$ bound itself. Nevertheless, even under a fixed budget, to the best of our knowledge, Theorem 3.8 still provides the first convergence guarantee for sparse ZO fine-tuning with a **Fisher-diagonal-guided, time-varying sampling rule**, rather than a fixed sparse subset.
>
> > Weakness 3
>
> We add new ablations on **OPT-2.7B** for w/o **EMA** and for comparing **$\sqrt{S}$-based sampling** (the default in CurvZO) with **$S$-based sampling**, as shown below. We also  report **OPT-6.7B** results with larger training sets (**2k** and **3k** examples); see our response to **Key Question 3** from reviewer **x5cB**. The ablations on **adaptive budget vs. fixed budget** and **curvature-guided sampling vs. random sampling** are reported in Table 4 of the **Appendix**.
>
> | Task                       |    SST-2 |      RTE |       CB |    BoolQ |      WSC |      WIC |    SQuAD |     DROP |
> | -------------------------- | -------: | -------: | -------: | -------: | -------: | -------: | -------: | -------: |
> | CurvZO(w/o EMA)            |     90.3 |     59.9 |     67.8 |     67.5 |     51.6 |     60.0 |     77.4 |     25.4 |
> | CurvZO($S$-based sampling) |     92.8 | **70.8** |     69.6 |     61.5 |     61.5 |     58.4 |     81.2 |     25.0 |
> | CurvZO                     | **94.1** |     67.7 | **69.8** | **68.9** | **62.9** | **61.9** | **81.5** | **27.6** |
>
> | Task                            |    SST-2 |      RTE |       CB |    BoolQ |      WSC |      WIC |    SQuAD |     DROP |
> | ------------------------------- | -------: | -------: | -------: | -------: | -------: | -------: | -------: | -------: |
> | CurvZO(w/o EMA)+LoRA            |     92.3 |     56.3 |     66.0 |     64.3 |     50.9 |     55.8 |     79.7 |     24.4 |
> | CurvZO($S$-based sampling)+LoRA |     93.5 |     57.4 |     60.7 |     65.1 |     60.3 |     56.8 |     78.9 |     24.4 |
> | CurvZO+LoRA                     | **93.6** | **64.7** | **69.6** | **66.1** | **63.4** | **57.6** | **81.1** | **27.6** |
>
> #

---

> > ### Author Rebuttal · Reviewer_VPph · 2026-04-02
> >
> > The rebuttal helps on the block-wise case and adds useful ablations. However, my main concern is only partially resolved. The practical method uses a normalized and EMA-smoothed curvature score, but the rebuttal still does not fully characterize its gap from the Fisher-diagonal-style target used in the theory. My overall evaluation stays the same.

---

> > > ### Author Response · Authors · 2026-04-07
> > >
> > > Thank you for the thoughtful follow-up.
> > > We agree that the remaining gap between the practical stabilized score and the Fisher-diagonal-style target is not yet fully characterized. This is challenging because normalization introduces a random denominator and EMA makes the score history-dependent. However, CurvZO does not require exact recovery of the Fisher diagonal. The score is only used to guide the **relative allocation of sampling probability across coordinates**, i.e., how the limited perturbation budget should be distributed. In this sense, the normalized and EMA-smoothed score is sufficient, since it provides a stable signal of relative coordinate importance for the sampling rule. While a tighter characterization would strengthen the theory, we believe the current theoretical result remains meaningful: to the best of our knowledge, this is the first convergence guarantee for sparse ZO fine-tuning with a Fisher-diagonal-guided, time-varying sampling rule.

---

### Official Review · Reviewer_x5cB · 2026-03-15

**Soundness:** 3
**Presentation:** 3
**Significance:** 3
**Originality:** 3
**Overall Recommendation:** 3
**Confidence:** 3

**Summary:**

The paper proposes CurvZO, a sparse zeroth-order (ZO) fine-tuning framework for large language models that learns an online, curvature-guided sampling distribution from scalar ZO feedback. While standard ZO is efficient because it circumvents backpropagation, it suffers from high-variance gradient estimates and slow convergence. CurvZO addresses this by tracking a local "curvature score" to identify the most informative parameters. It then applies a Horvitz-Thompson correction to maintain an unbiased gradient estimator and constructs a variance-minimizing sampling distribution that concentrates random perturbations on parameters with higher curvature. Additionally, it features an adaptive budget selection mechanism that dynamically scales the number of perturbed parameters based on the effective support size and sharpness of the evolving curvature distribution. Empirical evaluations demonstrate that CurvZO outperforms existing ZO baselines in both accuracy and convergence speed.

**Compliance With Llm Reviewing Policy:**

Affirmed.

**Key Questions For Authors:**

1.	Could you report ablations quantifying the contribution of each component: (i) curvature guidance vs uniform/random π, (ii) π ∝ √S vs π ∝ S or top-k, (iii) normalized vs unnormalized scores, (iv) EMA on/off, and (v) adaptive budget vs fixed budget?
2.	What is the overhead (time and memory) of maintaining block-wise scores for Llama2-13B? How many blocks are used, and how often are π and B recomputed?
3.	How stable is CurvZO under longer training or larger data regimes (>1k examples)? Do gains persist or diminish as the task data grows?

**Limitations:**

Yes.

**Strengths And Weaknesses:**

Strengths:

1.	The theoretical foundation is rigorous. The derivation linking the scalar-derived curvature score to the diagonal Fisher information matrix is mathematically sound. The use of the Horvitz-Thompson reweighting to ensure the sparse gradient estimator remains unbiased is a necessary choice for ensuring convergence.

2.	The presentation is well-structured and easy to follow, code is good and clear.
3.	Overcoming the memory wall in LLM fine-tuning is a highly relevant and pressing problem in the field. CurvZO advances the capabilities of ZO optimization by significantly reducing the iterations required to converge while maintaining the core benefit of avoiding intermediate activation storage.

Weaknesses:

1.	Variance analysis relies on approximations that conflate global terms into a constant C and omit potentially important dependence on masking-induced correlations; the bound Var(ĝ_i) ≤ C F_ii/π_i is plausible but somewhat hand-wavy in its current form. (65->66)
2.	Missing comparisons to several relevant ZO fine-tuning baselines that specifically target sparsity/structure like MeZO-BCD (recent adaptive block-selection ZO methods). Sparse-MeZO, SubZero (random subspaces), SensZOQ are discussed but not compared empirically.
3.	Limited ablation in the main text: the incremental benefit of each component (score normalization, EMA, sqrt(S) vs S sampling, and budget adaptation vs fixed budget) is not shown.

---

> ### Author Rebuttal · Authors · 2026-03-31
>
> We thank the reviewer for the valuable suggestions.
>
> > Weakness 1
>
> We agree that the approximation from Eq. (65) to Eq. (66) is not explained clearly enough, but it is well motivated for deriving the sampling rule. Specifically, Eq. (65) gives
> $$
> \mathrm{Var}(\tilde{g}\_i)\leq\frac{3}{\pi_i}F_{ii}+\frac{1}{\pi_i}\sum_{j\neq i}\pi_jF_{jj}\leq\frac{3F_{ii}+\sum_j\pi_jF_{jj}}{\pi_i}.
> $$
> This bound contains: (i) a coordinate-wise term $3 F_{ii}/\pi_i$, which captures the Fisher-dependent contribution of coordinate $i$ and (ii) a shared term $\sum_j \pi_j F_{jj}/\pi_i$, which depends on a global Fisher aggregate.
>
> Our choice to focus on the coordinate-wise term and absorb the shared term into a constant is motivated by:
>
> - **Sampling motivation:** the coordinate-wise term determines how sampling probabilities should vary across coordinates, which is exactly what CurvZO needs to allocate a limited perturbation budget via the sampling rule.
> - **Online tractability:** retaining the shared term leads to a coupled optimization problem without a simple closed-form solution, making the rule impractical for online use.
>
> Since ZO fine-tuning uses noisy online curvature scores rather than exact Fisher diagonals, we adopt the simplified form to preserve the main coordinate-wise dependence while keeping the rule tractable.
>
> > Weakness 2
>
> We include the following additional ZO baselines on OPT-2.7B. SensZOQ and MeZO-BCD do not support LoRA.
>
> | Task        | SST-2    | RTE      | CB       | BoolQ    | WSC      | WIC      | SQuAD    | DROP     |
> | ----------- | -------- | -------- | -------- | -------- | -------- | -------- | -------- | -------- |
> | Sparse-MeZO | 92.6     | 66.8     | 69.6     | 62.8     | 61.5     | 55.2     | 79.6     | 24.9     |
> | SensZOQ     | 90.5     | 60.6     | 64.2     | 64.8     | 54.2     | 55.8     | 72.9     | 25.1     |
> | MeZO-BCD    | 93.5     | 64.9     | 67.8     | 66.0     | 57.6     | 57.0     | 80.4     | 23.5     |
> | SubZero     | 85.8     | 54.4     | **71.0** | 65.3     | 60.5     | 52.5     | 61.7     | 18.4     |
> | CurvZO      | **94.1** | **67.7** | 69.8     | **68.9** | **62.9** | **61.9** | **81.5** | **27.6** |
>
> | Task             |    SST-2 |      RTE |       CB |    BoolQ |      WSC |      WIC |    SQuAD |     DROP |
> | ---------------- | -------: | -------: | -------: | -------: | -------: | -------: | -------: | -------: |
> | Sparse-MeZO+LoRA |     93.1 |     57.4 |     62.5 |     65.2 |     53.8 |     54.2 |     80.8 |     24.4 |
> | SubZero+LoRA     |     82.6 |     54.2 |     59.1 |     64.8 |     51.4 |     53.2 |     62.4 |     17.2 |
> | CurvZO+LoRA      | **93.6** | **61.7** | **69.6** | **66.1** | **63.4** | **57.6** | **81.1** | **27.6** |
>
> The results show that CurvZO generally outperforms the new baselines in both settings.
>
> > Weakness 3 and Key Question 1
>
> Table 4 in the Appendix reports ablations on **adaptive budget selection vs. fixed budget**, and **curvature-guided sampling vs. random sampling**. We further add new ablations on **OPT-2.7B** by removing **EMA** and by comparing **$\sqrt{S}$-based sampling** (the default in CurvZO) with **$S$-based sampling**; see our response to **Weakness 3** from reviewer **VPph**. We do not include an ablation without **score normalization**, because removing normalization causes much larger score fluctuations, unstable training, and worse performance.
>
> > Key Question 2
>
> In our implementation, each block corresponds to a trainable parameter tensor (e.g., a projection matrix or a bias vector). For Llama2-13B, this yields **363 blocks**. Storing one scalar score per block requires only **363 float32 values** (about **1.4 KB**), which is negligible relative to the model size. We update $\pi$ **at every step**, at an average cost of **0.36 s** per update, while $B$ is updated **every 20 steps**, costing only **0.000193 s** per update.
>
> > Key Question 3
>
> Following prior ZO fine-tuning work, our main experiments use the standard **1k-example** setting for direct comparison. To address the reviewer’s question, we additionally report results on **OPT-6.7B** with larger training sets (**2k** and **3k** examples), as shown below. The results indicate that CurvZO remains effective beyond the 1k-example regime.
>
> | Task        |    SST-2 |      RTE |      WIC |      WSC |    SQuAD |
> | ----------- | -------: | -------: | -------: | -------: | -------: |
> | CurvZO (1k) |     94.7 |     72.2 |     61.7 |     61.5 |     83.7 |
> | CurvZO (2k) | **94.8** |     73.8 |     63.1 |     63.4 |     84.2 |
> | CurvZO (3k) |     94.5 | **79.0** | **64.4** | **63.6** | **84.9** |
>
> | Task             |    SST-2 |      RTE |      WIC |      WSC |    SQuAD |
> | ---------------- | -------: | -------: | -------: | -------: | -------: |
> | CurvZO+LoRA (1k) |     93.1 |     63.5 | **61.5** |     63.4 |     80.6 |
> | CurvZO+LoRA (2k) | **93.6** |     64.2 |     59.0 | **64.2** | **80.9** |
> | CurvZO+LoRA (3k) |     93.3 | **64.6** |     59.9 |     62.4 |     80.3 |
>
> #

---

> > ### Author Rebuttal · Reviewer_x5cB · 2026-04-03
> >
> > Thanks the authors for their response, which addresses part of our concerns.
> >
> > I agree with the reviewers that the empirical evaluation of this paper is weak.

---

> > > ### Author Response · Authors · 2026-04-07
> > >
> > > Thank you for the follow-up. To address the concern regarding the empirical evaluation, we have further strengthened the experiments by adding **runtime comparisons**, evaluating a newer model (**Llama3.1-8B**), and including more challenging benchmarks (**Dolly** and **GSM8K**) (please see our response to **Reviewer 8aw8** for details). We hope these additions provide a more comprehensive empirical validation of **CurvZO**.
> > >
> > > More specifically, **CurvZO** is designed as a **memory-efficient fine-tuning method for LLMs under resource-constrained settings**, where standard backpropagation-based fine-tuning is often impractical due to its high memory overhead. The main contributions of this paper are:
> > >
> > > * an **online curvature-tracking mechanism** derived from scalar ZO feedback;
> > > * a **curvature-guided sparse sampling rule** together with **adaptive budget selection**;
> > > * corresponding **theoretical analysis**, including a convergence guarantee for **Fisher-diagonal-guided, time-varying sparse ZO updates**;
> > > * extensive empirical results demonstrating improved accuracy, faster convergence, and strong memory efficiency.

---

### Decision · Program_Chairs · 2026-04-30

**Decision:**

Accept (regular)

**Comment:**

Reviewers confirmed that the rebuttal effectively addressed their concerns and acknowledged the paper's novel approach to an important problem. However, several reviewers noted post-rebuttal that experimental evidence only partially supports the claims. Specifically, performance gaps emerge on more challenging benchmarks compared to competing methods. Despite these limitations, reviewers recognized that the methodological contributions merit publication if there is room in the program and will be of value to the community.